# SCORE MATCHING VIA DIFFERENTIABLE PHYSICS

## ABSTRACT

Diffusion models based on stochastic differential equations (SDEs) gradually perturb a data distribution $p(\mathbf{x})$ over time by adding noise to it. A neural network is trained to approximate the score $\nabla_{\mathbf{x}} \log p_t(\mathbf{x})$ at time $t$, which can be used to reverse the corruption process. In this paper, we focus on learning the score field that is associated with the time evolution according to a physics operator in the presence of natural non-deterministic physical processes like diffusion. A decisive difference to previous methods is that the SDE underlying our approach transforms the state of a physical system to another state at a later time. For that purpose, we replace the drift of the underlying SDE formulation with a differentiable simulator or a neural network approximation of the physics. At the core of our method, we optimize the so-called probability flow ODE to fit a training set of simulation trajectories inside an ODE solver and solve the reverse-time SDE for inference to sample plausible trajectories that evolve towards a given end state. We demonstrate the competitiveness of our approach for different challenging inverse problems.

## 1   INTRODUCTION

Many physical systems are time-reversible on a microscopic scale. For example, a continuous material can be represented by a collection of interacting particles (Gurtin, 1982; Blanc et al., 2002) based on which we can predict future states of the material. We can also compute earlier states, meaning we can evolve the simulation backwards in time (Martyna et al., 1996). When taking a macroscopic perspective, we only know the average quantities within specific regions (Farlow, 1993), which constitutes a loss of information. It is only then that time-reversibility is no longer possible, since many macroscopic and microscopic initial states exist that evolve to yield the same macroscopic state.

In the following, we target inverse problems to reconstruct the distribution of initial macroscopic states for a given end state. This genuinely tough problem has applications in many areas of scientific machine learning (Zhou et al., 1996; Gómez-Bombarelli et al., 2018; Delaquis et al., 2018; Lim & Psaltis, 2022), and existing methods lack tractable approaches to represent and sample the distribution of states. We address this issue by leveraging continuous approaches for diffusion models in the context of physical simulations. In particular, our work builds on the *reverse-diffusion* theorem (Anderson, 1982). Given the functions $f(\cdot, t) : \mathbb{R}^d \rightarrow \mathbb{R}^d$, called *drift*, and $g(\cdot) : \mathbb{R} \rightarrow \mathbb{R}$, called *diffusion*, it can be shown that under mild conditions, for the forward stochastic differential equation (SDE) $d\mathbf{x} = f(\mathbf{x}, t)dt + g(t)dw$ there is a corresponding reverse-time SDE $d\mathbf{x} = [f(\mathbf{x}, t) - g(t)^2 \nabla_{\mathbf{x}} \log p_t(\mathbf{x})]dt + g(t)d\tilde{w}$. In particular, this means that given a marginal distribution of states $p_0(\mathbf{x})$ at time $t = 0$ and $p_T(\mathbf{x})$ at $t = T$ such that the forward SDE transforms $p_0(\mathbf{x})$ to $p_T(\mathbf{x})$, then the reverse-time SDE runs backward in time and transforms $p_T(\mathbf{x})$ into $p_0(\mathbf{x})$. The term $\nabla_{\mathbf{x}} \log p_t(\mathbf{x})$ is called the *score*.

This theorem is a central building block for SDE-based diffusion models and denoising score matching (Song et al., 2021c; Jolicoeur-Martineau et al., 2021), which parameterize the drift and diffusion in such a way that the forward SDE corrupts the data and transforms it into random noise. By training a neural network to represent the score, the reverse-time SDE can be deployed as a generative model, which transforms samples from random noise $p_T(\mathbf{x})$ to the data distribution $p_0(\mathbf{x})$.

In this paper, we show that a similar methodology can likewise be employed to model physical processes. We replace the drift $f(\mathbf{x}, t)$ by a physics model $\mathcal{P}(\mathbf{x}) : \mathbb{R}^d \rightarrow \mathbb{R}^d$, which is implemented by a differentiable solver or a neural network that represent the dynamics of a physical system, thus

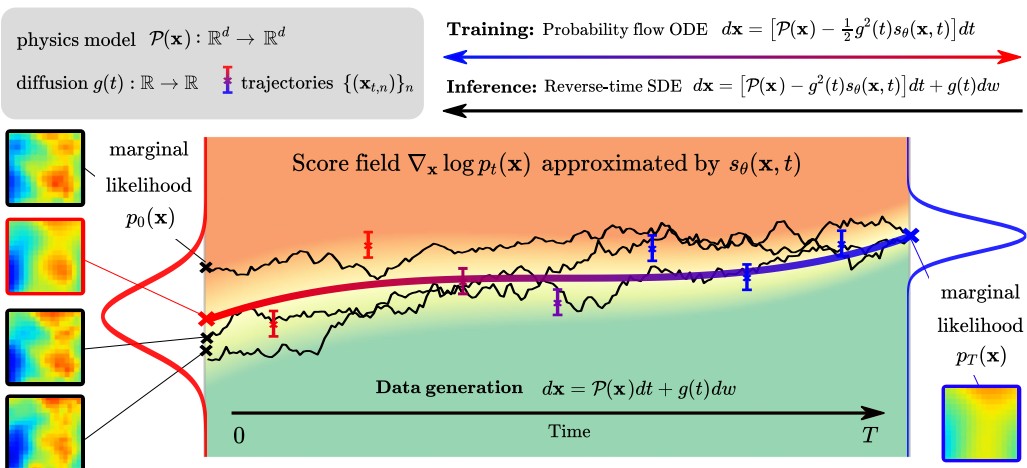

**Figure 1:** Overview: we employ a physics simulator $\mathcal{P}$ to learn the score field $\nabla_x \log p_t(\mathbf{x})$ with a neural network $s_\theta$ in the presence of noise or uncertainties. The trained model allows for sampling the posterior of $p_0$, i.e. different states that explain an observation $p_T$, via probability flow or by solving the reverse-time SDE.

deeply integrating physical knowledge into our method. The end state at $t = T$ on which the forward SDE acts is not fully destroyed by the diffusion $g(t)$, but instead, the noise acts as a perturbation of the system state over time. An overview of our method is shown in figure 1.

To the best of our knowledge, our work is the first to leverage the reverse-diffusion theorem as a method for solving inverse problems of physical systems. As such, our primary aim is to demonstrate how existing algorithms from this field can be used in the context of physics simulations. We showcase the efficacy of the score matching viewpoint on physics problems with a range of challenging inverse problems. Specifically, our contributions are:

1. We develop a framework in which we incorporate the reverse-diffusion theorem and score matching into a method for solving inverse problems that involve the time evolution of physical systems. We demonstrate its competitiveness against common baseline approaches using the heat equation as an example.

2. We highlight the effectiveness of our method with a more challenging inverse problem where we simulate a fluid-based transport process backwards in time in the presence of randomized obstacles. Here, we compare our method to different strategies for learned solvers.

3. Finally, we show that this approach can even be used when the underlying SDE is unknown. Our approach can be combined with operator learning methods and we demonstrate its effectiveness for learning the Navier-Stokes equation in the turbulent regime.

## 2 BACKGROUND AND RELATED WORK

*Learned solvers:* Numerical simulations benefit greatly from machine learning models (Tompson et al., 2017; Morton et al., 2018; Pfaff et al., 2020; Li et al., 2020). By integrating a neural network inside differential equation solvers, it is possible learn to reduce numerical errors (Tompson et al., 2017; Kochkov et al., 2021; Brandstetter et al., 2022) or guide the simulation towards a desired target state (Holl et al., 2020b; Li et al., 2022). As errors may accumulate quickly over time, trained networks benefit from gradients that are backpropagated over multiple time steps (Um et al., 2020).

*Diffusion models:* Diffusion models (Ho et al., 2020; Song et al., 2021c) have been considered for a wide range of applications. Most notably, diffusion models have been proposed for image (Dhariwal & Nichol, 2021), video (Ho et al., 2022; Höppe et al., 2022; Yang et al., 2022) and audio synthesis (Chen et al., 2021). Recently, Bansal et al. (2022) have proposed to train generalized diffusion models for arbitrary transformations and suggest that fully deterministic models without any noise are sufficient for generative behaviour.

Specifically for uncertainty quantification, solving inverse problems and conditional sampling many methods have been proposed (Chung et al., 2021; 2022; Song et al., 2021b; Kawar et al., 2021; Ramzi et al., 2020). However, most approaches either focus on the denoising objective that is common for tasks involving natural images, or the synthesis process of solutions does not take the underlying physics directly into account.

*Generative modeling via SDEs:* Classical denoising score matching approaches based on Langevin dynamics (Vincent, 2011; Song & Ermon, 2019, SMLD) and based on discrete Markov chains, e.g. Denoising Diffusion Probabilistic Models (Sohl-Dickstein et al., 2015; Ho et al., 2020, DDPM), can be unified in a time-continuous framework using SDEs (Song et al., 2021c). Given a distribution of states $p_0(\mathbf{x})$ at time $t = 0$, an SDE transforms $p_0(\mathbf{x})$ to a tractable distribution $p_T(\mathbf{x})$

$$d\mathbf{x} = f(\mathbf{x}, t)dt + g(t)dw, \tag{1}$$

with $w$ the standard Brownian motion, a *drift* $f(\cdot, t) : \mathbb{R}^d \to \mathbb{R}^d$ and *diffusion* $g(\cdot) : \mathbb{R} \to \mathbb{R}$, which for $\mathbf{x}_0 \sim p_0(\mathbf{x})$ yields a diffusion process $(\mathbf{x}_t)_{t=0}^T$. As outlined above, the reverse-time SDE of the reverse-diffusion theorem (Anderson, 1982) is given by

$$d\mathbf{x} = [f(\mathbf{x}, t) - g(t)^2 \nabla_{\mathbf{x}} \log p_t(\mathbf{x})]dt + g(t)d\tilde{w}, \tag{2}$$

with $\nabla_{\mathbf{x}} \log p_t(\mathbf{x})$ being the *score*. By sampling from the tractable distribution $p_T(\mathbf{x})$ and simulating the reverse-time SDE equation (2), we can generate samples from $p_0(\mathbf{x})$. Since the score $\nabla_{\mathbf{x}} \log p_t(\mathbf{x})$ is not known analytically, it is approximated by a neural network.

*Continuous normalizing flows:* Continuous normalizing flows (CNFs) are invertible generative models based on neural ODEs. Given an initial distribution $z_0 \sim p_{z_0}(z_0)$ and a function $f(z(t), t; \theta)$ represented by a neural network, a CNF is trained to solve the ODE $\partial z(t)/\partial t = f(z(t), t; \theta)$ with boundary conditions $z(t_0) = z_0$ and $z(t_1) = \mathbf{x}$, where $\mathbf{x} \in \mathbb{R}^d$ is a sample from the training data set. A useful property of CNFs is that it is cheap to compute the log-likelihood of data samples, due to the instantaneous change of variables formula $\partial \log p(z(t))/\partial t = -\text{Tr}(\partial f/\partial z(t))$ (Grathwohl et al., 2019). The evolution of the marginal probability density $p_t(\mathbf{x})$ for the SDE in equation (1) is described by Kolmogorov's forward equation (Øksendal, 2003). Maoutsa et al. (2020) and Song et al. (2021c) show that there exists an ODE with the same Kolmogorov forward equation. This ODE is called *probability flow* ODE and is given by

$$d\mathbf{x} = \left[ f(\mathbf{x}, t) - \frac{1}{2} g(t)^2 \nabla_{\mathbf{x}} \log p_t(\mathbf{x}) \right] dt. \tag{3}$$

The probability flow ODE equation (3) represents a CNF and, if $f(\mathbf{x}, t)$ is known, a network $s_\theta$ parameterized by $\theta$ representing $\nabla_{\mathbf{x}} \log p_t(\mathbf{x})$ can be trained via maximum likelihood using standard methods (Chen et al., 2018). While the evolution of $p_t(\mathbf{x})$ is the same between the probability flow ODE from equation (3) and the reverse-time SDE from equation (2), there are caveats due to the approximation by $s_\theta(\mathbf{x}, t)$ (Song et al., 2021b; Lu et al., 2022). Huang et al. (2021) show that minimizing the score-matching loss is equivalent to maximizing a lower bound of the likelihood obtained by sampling from the reverse-time SDE.

A recent variant combines score matching with CNFs (Zhang & Chen, 2021), and employs a joint training of the drift and score with an integration backwards in time.

## 3 METHOD

**Modeling assumptions** We consider a known physics model $\mathcal{P}(\mathbf{x}) : \mathbb{R}^d \to \mathbb{R}^d$ that is differentiable and approximates the time evolution, i.e. $\mathbf{x}_{t_{m+1}} \approx \mathbf{x}_{t_m} + (t_{m+1} - t_m) \cdot \mathcal{P}(\mathbf{x}_{t_m})$. One of our key modelling choices is to describe the time evolution of the physical system by a stochastic differential equation

$$d\mathbf{x} = \mathcal{P}(\mathbf{x})dt + g(t)dw, \tag{4}$$

with a diffusion process $g(\cdot) : \mathbb{R} \to \mathbb{R}$ that perturbs the simulation states. We can simulate paths from this SDE using Euler-Maruyama steps, i.e. for a time discretization $t_0 < t_1 < ... < t_M$ and initial state $\mathbf{x}_{t_0}$, we obtain the iteration rule

$$\mathbf{x}_{t_{m+1}} \leftarrow \mathbf{x}_{t_m} + (t_{m+1} - t_m) \cdot \mathcal{P}(\mathbf{x}_{t_m}) + \sqrt{t_{m+1} - t_m} \cdot g(t_m) z_{t_m}, \tag{5}$$

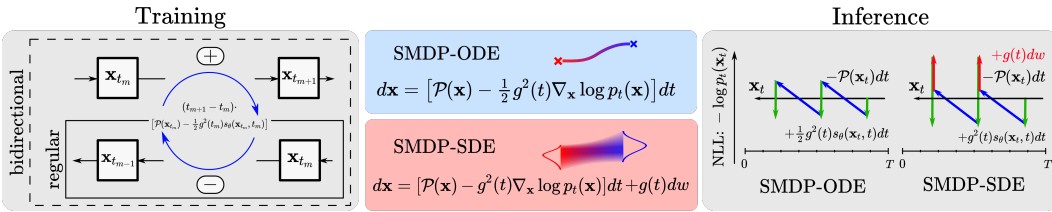

**Figure 2:** During the training phase, we optimize $s_\theta(\mathbf{x}, t)$ that approximates the score $\nabla_\mathbf{x} \log p_t(\mathbf{x})$ inside the probability flow ODE to fit the data trajectories. For the inference part, we simulate trajectories of the reverse-time SDE.

where $z_{t_m}$ are i.i.d. with $z_{t_m} \sim \mathcal{N}(0, I)$. The random noise $g(t_m)z_{t_m}$ that is added to the data at each time step can be regarded as either noise inherent to the physical problem, or as noise from a measurement process. As training data, we consider a set of $N$ trajectories $\{(\mathbf{x}_{t_i,n})_{i=0,...,M}\}_{n=0,...,N}$ sampled with equation (5) and which describe the evolution of a physical system.

**Inverse Problem** Given an end state $\mathbf{x}_T$ we are interested in recovering a likely trajectory $(\mathbf{x}_{t_i}^{\text{pred}})_{i=0,...,M}$ that evolves towards $\mathbf{x}_T$. More formally, the set of trajectories $\{(\mathbf{x}_{t_i,n})_{i=0,...,M}\}_{n=0,...,N}$ implicitly defines marginal likelihoods $p_t(\mathbf{x})$ at every time step $t$ which are linked through the SDE equation (4) of the physical system by the Kolmogorov forward equation. The solution trajectory may not be unique, so we want to sample from the full posterior instead of obtaining only a maximum likelihood solution, i.e. we want to sample from $p_0(\mathbf{x}|\mathbf{x}_T)$.

**Method** In line with previous work in score-based generative modeling (Song & Ermon, 2019; Song et al., 2021c), we approximate the score $\nabla_\mathbf{x} \log p_t(\mathbf{x})$ of the marginal likelihoods by a neural network $s_\theta(\mathbf{x}, t)$. We optimize $s_\theta(\mathbf{x}, t)$ via maximum likelihood training of the probability flow ODE, as discussed in section 2. For this, we maximize a variational lower bound for the maximum likelihood objective, which we estimate by minimizing the following loss

$$\mathcal{L}\left((\mathbf{x}_{t_i})_{i=0}^M, \theta\right) = \sum_{i=1}^M ||\mathbf{x}_{t_i} - \mathbf{x}_{t_i}^{\text{ODE}}||_2^2 \tag{6}$$

$$\text{s.t.} \quad \mathbf{x}_0^{\text{ODE}} = \mathbf{x}_T + \int_T^0 \mathcal{P}(\mathbf{x}_t^{\text{ODE}}) - \frac{1}{2}g^2(t)s_\theta(\mathbf{x}_t^{\text{ODE}}, t)dt, \tag{7}$$

where we sample a SDE trajectory $(\mathbf{x}_{t_i})_{i=0}^M$ from the training set. We give theoretical justification for this objective in appendix A. Intuitively, our method fits bijective and deterministic trajectories of the probability flow ODE to the non-deterministic SDE trajectories. In contrast to previous work, our method deeply integrates a prior about the physical system in the form of the simulation operator $\mathcal{P}(\mathbf{x})$ into the training process. In this context, the end state at $t = T$ is not fully destroyed by the noise, but instead the noise acts as a perturbation of the system state over time. An overview of our method is shown in figure 2.

Given an end state $\mathbf{x}_T$, we can solve the probability flow ODE backwards in time using the trained score function $s_\theta(\mathbf{x}, t)$ to obtain a trajectory $(\mathbf{x}_{t_i}^{\text{pred}})_{i=0,...,M}$. However, this will only give a single, deterministic solution and not allow for sampling from the posterior $p(\mathbf{x}|\mathbf{x}_T)$. We simulate trajectories from the reverse-time SDE (see section 1) via

$$d\mathbf{x} = \left[\mathcal{P}(\mathbf{x}) - g^2(t)s_\theta(\mathbf{x}, t)\right] dt + g(t)dw. \tag{8}$$

The evolution of marginal probabilities $p_t(\mathbf{x})$ for this SDE is the same as for the probability flow ODE equation (7) (Song et al., 2021c). Moreover, by the reverse-diffusion theorem (Anderson, 1982), SDE equation (8) is the time-reverse of the physical system SDE from equation (4). Therefore, we can approximate sampling from $p(\mathbf{x}|\mathbf{x}_T)$ by simulating trajectories from SDE equation (8) using any traditional SDE solver. In the following, we refer to the integration of the physics model $\mathcal{P}(\mathbf{x})$ into the score-based modelling approach as *score matching via differentiable physics*, or *SMDP* in short. We denote trajectories from the probability flow ODE by *SMDP-ODE*, and those obtained by simulating the reverse-time SDE by *SMDP-SDE*.

**Training and Inference**  Algorithm 1 gives an overview of SMDP inference for the ODE as well as the SDE variant when using the explicit Euler method as ODE solver. For simplicity, we also employ the explicit Euler method for training and backpropagate gradients through multiple solver steps when computing the ODE trajectory in equation (7) to obtain updates for $\theta$. We also refer to this procedure as unrolling the dynamics. Our training setup is similar to Um et al. (2020), which was originally developed for training correction functions in the context of controlling numerical errors for physical simulations. In particular, in our implementa-

---

**Algorithm 1** SMDP-ODE, SMDP-SDE

**Require:** $\mathbf{x}_{t_M}$, $\{t_m\}_{m=0}^{M}$, $\{g_{t_m}\}_{m=0}^{M}$
1: **for** $m \leftarrow M$ to $1$ **do**
2:     $\mathbf{p} \leftarrow \mathcal{P}(\mathbf{x}_{t_m})$
3:     $\mathbf{s} \leftarrow -g_{t_m}^2 s_\theta(\mathbf{x}_{t_m}, t_m)/2$
4:     **if** SMDP-ODE **then**
5:         $\mathbf{x}_{t_{m-1}} \leftarrow \mathbf{x}_{t_m} - (t_m - t_{m-1}) \cdot (\mathbf{p} + \mathbf{s})$
6:     **if** SMDP-SDE **then**
7:         $\mathbf{x}_{t_{m-1}} \leftarrow \mathbf{x}_{t_m} - (t_m - t_{m-1}) \cdot (\mathbf{p} + 2\mathbf{s})$
8:         $z \sim \mathcal{N}(0, I)$
9:         $\mathbf{x}_{t_{m-1}} \leftarrow \mathbf{x}_{t_{m-1}} + g_{t_m}\sqrt{t_m - t_{m-1}}z$
10: **return** $\mathbf{x}_{t_0}$

---

tion, we consider a sliding window for unrolling the dynamics, which makes our training very flexible, i.e. we can consider single-step updates as well as unrolling the entire simulation. We give a more detailed descriptions about our training setup in appendix A. We consider an additional variant of SMDP, for which we apply a *bidirectional training*, i.e. instead of only training the probability flow ODE for the time-backward direction $T \rightarrow 0$ we alternate with optimizing the time-forward direction $0 \rightarrow M$, i.e. equation (7) becomes

$$\mathbf{x}_T^{\mathrm{ODE}} = \mathbf{x}_0 + \int_0^T \mathcal{P}(\mathbf{x}_t^{\mathrm{ODE}}) - \frac{1}{2}g^2(t)s_\theta(\mathbf{x}_t^{\mathrm{ODE}}, t)dt, \tag{9}$$

## 4 EXPERIMENTS

We conduct several experiments to demonstrate the advantages of SMDP compared to a number of baseline methods. The source code for all experiments will be made available upon acceptance. We first consider the 2D heat equation in section 4.1, where our objective is to find possible initial states at time $t = 0$ given a noisy end state at time $t = T$.

In our second experiment in section 4.2, we transfer the established practices to a more challenging problem, where we are interested in reconstructing the trajectory of a buoyancy-driven flow within a closed simulation domain given an end state at time $T$. What makes this problem challenging is that for each simulation, we place different obstacles at different positions within the simulation domain. Then, in section 4.3, we consider the situation where the physics of the system is unknown. For this purpose, we consider training a network that approximates the solutions to the Navier-Stokes equations and a network that approximates the score field. We demonstrate that by doing so, we obtain an improved performance for inverse problems and the learned score can be used to refine predictions in post-processing. We provide additional results in the appendix. In appendix B we compare our method with implicit score matching (Hyvärinen, 2005) in a toy experiment, which demonstrates the importance of include physics dynamics in the training. We analyze how many steps are required when unrolling the dynamics for obtaining stable trajectories in appendix C. Finally, in appendix we give an additional evaluation of quality and diversity of the posterior distribution we obtain when sampling from the reverse-time SDE in the heat equation experiment.

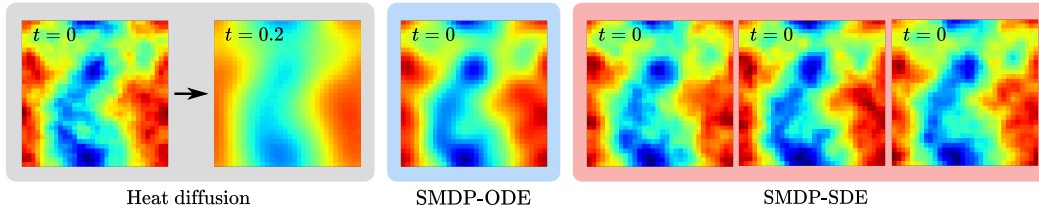

**Figure 3:** Heat diffusion case. We simulate a Gaussian random field at $t = 0$ forwards in time using equation (5). Given $s_\theta$ we can either solve the probability flow ODE or simulate trajectories of the reverse-time SDE to obtain solutions for the state at $t = 0$.

| Method | MSE $[10^{-5}]\downarrow$ | Spectral error $\downarrow$ | Full posterior |
|---|---|---|---|
| SMDP-ODE | **0.74** | 3.62 | ✗ |
| SMDP-SDE | 5.56 | **0.56** | ✓ |
| | | | |
| ResNet-S | 2.17 | 1.67 | ✗ |
| ResNet-P | 2.30 | 1.09 | ✗ |
| BNN-S | $3.47 \times 10^2$ | 1.25 | ✓ |
| BNN-P | $3.81 \times 10^2$ | 0.99 | ✓ |
| FNO-S | $2.54 \times 10^4$ | 1.60 | ✗ |
| FNO-P | $2.50 \times 10^4$ | 1.47 | ✗ |
| HeatGen | 1.39 | 4.45 | ✗ |
| HeatGen+noise | 4.45 | 3.24 | ✓ |

**Table 1:** Evaluation of reconstruction MSE and spectral error for SMDP and baselines. The column full posterior indicates whether models yield point estimates or allow to sample from the posterior.

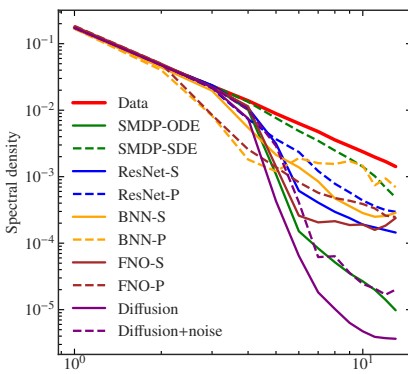

**Figure 4:** Spectral density on different scales, the red line indicating ground truth. The closer a method is to the ground truth, the better it produces structures of a similar scale.

## 4.1 HEAT EQUATION

We consider the heat equation $\frac{\partial u}{\partial t} = \alpha \Delta u$ which plays a fundamental role in many physical systems. Here, we set the diffusivity constant to $\alpha = 1$ and initial conditions at $t = 0$ are generated from Gaussian random fields with $n = 4$ at resolution $32 \times 32$. We simulate the heat diffusion using a spectral method until $t = 0.2$ with a fixed step size $\Delta t = 6.25 \times 10^{-3}$ using the iteration rule from equation (5) with $g \equiv 0.1$. Our training data set consists of 2.500 initial conditions with their corresponding trajectories sampled with varying step size $\Delta t$ and end states at $t = 0.2$. The test set is comprised of 500 initial conditions and corresponding end states generated directly without any noise.

**Training** We consider a small *ResNet*-like architecture based on an encoder and decoder part (see appendix E) as representation for the score function $s_\theta(\mathbf{x}, t)$. The physics model $\mathcal{P}$ is implemented via differentiable programming in *JAX* (Schoenholz & Cubuk, 2020). For better comparison with the baseline methods, these are trained with a Gaussian random noise of $\sigma = 0.1$ added to the inputs. This noise is applied to all network inputs during testing.

**Baseline methods** As baseline methods, we consider the ResNet-like architecture from above, in addition to a *Bayesian neural network* (BNN) based on a U-Net architecture with spatial dropout (Mueller et al., 2022), as well as a *Fourier neural operator* (FNO) network (Li et al., 2020). For each of these three methods, we consider two variants: the first variant is trained with a *supervised loss*, i.e. the training data consists of pairs $(\mathbf{x}_0, \mathbf{x}_T)$ with initial state $\mathbf{x}_0$ and end state $\mathbf{x}_T$. The supervised loss corresponds to the squared L2 distance between the network prediction $\mathbf{x}_0^{\text{pred}}$ and the ground truth, i.e. $(\mathbf{x}_0^{\text{pred}} - \mathbf{x}_0)^2$. For the second variant, the *reconstruction loss*, we rely on the differentiable solver and only make use of the end state $\mathbf{x}_T$ such that the loss becomes $(\mathcal{P}(\mathbf{x}_0^{\text{pred}}; T) - \mathbf{x}_T)^2$, i.e we simulate the network output forward in time using $\mathcal{P}$ to obtain a state at $t = T$, which we compare to the desired end state $\mathbf{x}_T$. We denote the supervised variant by *S* and the physics-based one by *P*. Additionally, we consider an adopted generative model from Rissanen et al. (2022), denoted by *HeatGen*. We train this network similarly to SMDP-ODE, but without the solver $\mathcal{P}$, such that the network has to learn the score and the physics at the same time.

**Reconstruction accuracy vs. fitting the data manifold** We give an evaluation of our method and the baselines by considering the *reconstruction MSE* on the test set: how well a predicted initial state $\hat{\mathbf{x}}_0$ that is simulated forward in time yields states that correspond to the reference end state $\mathbf{x}_T$ in terms of MSE. This metric has the disadvantage that it does not measure how well the prediction matches the training data manifold, i.e. for this case whether the prediction resembles the statistics of the Gaussian random field. For that reason, we additionally compare the power spectral density of the states as the *spectral loss*. The corresponding measurements are given in table 1, which

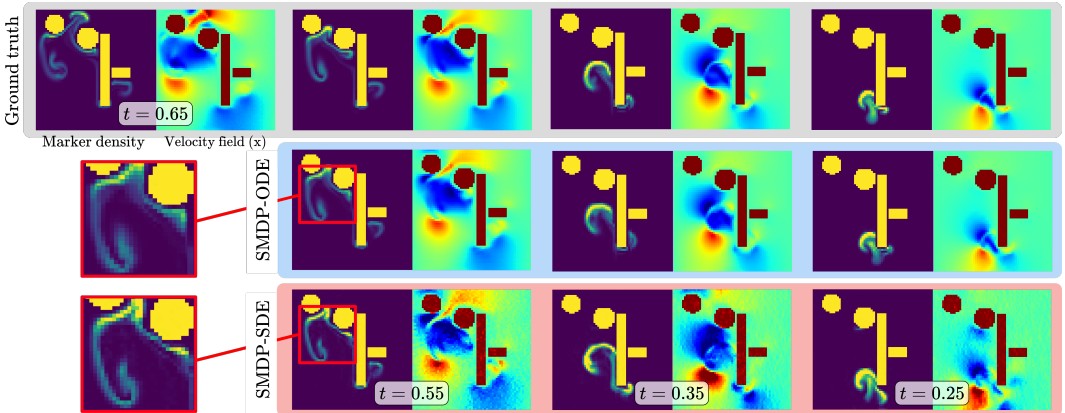

**Figure 6:** Buoyancy flow case. Ground truth shows the marker density and velocity field in the $x$-direction at different points of the simulation trajectories. The simulation end state at $t = 0.65$ is the input to SMDP-ODE and SMDP-SDE.

show that our method SMDP-ODE performs best in terms of the reconstruction MSE. However, solutions obtained by SMDP-ODE are very smooth and do not contain the small-scale structures of the references, which is reflected in a high spectral error that is also visually prominent, as shown in figure 4. SMDP-SDE on the other hand performs very well in terms of spectral error and yields visually convincing solutions with only a slight increase in the reconstruction loss. We note that there is a natural tradeoff between both metrics, and SMDP-ODE and SMDP-SDE perform best for both cases respectively while using the same set of weights.

**Ablation study** We performed an ablation study to highlight several design choices of the proposed method. In particular we note that despite fundamental differences between SMDP-ODE and SMDP-SDE, as explained in section 3, the main difference at inference time is the constant factor for $s_\theta$ and the noise term $g(t)dw$ for SMDP-SDE, cf. equations (7) and (8). We investigated how the change in noise integration affects the performance of SMDP-ODE, and considered a variant *SMDP-ODE+noise* that includes the addition of the noise term during inference, but is otherwise identical to SMDP-ODE. As shown with crosses in figure 5, this method has a slightly higher reconstruction loss but in contrast to SMDP-SDE does not improve upon the spectral error. This indicates that SMDP-SDE can recover small-scale distributions of the references, while the ODE variant by construction tends to favour smooth solutions when faced with uncertainties. We additionally investigated the effect of the proposed *bidirectional* training scheme. The error measurements of Figure 5 show that ODE variant is not affected, but SMDP-SDE significantly benefits. The resulting model robustly handles a wide range of different

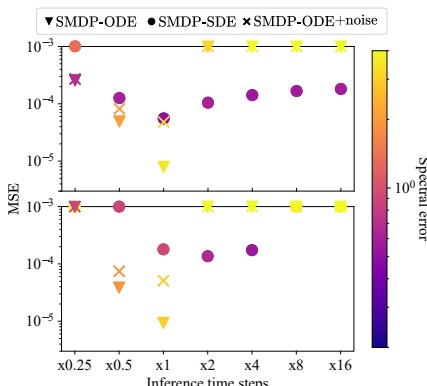

**Figure 5:** Reconstruction MSE and spectral errors for the bidirectional (top) and regular variant (bottom). The x-axis shows the relative increase of the number of time steps during inference compared to training. Large errors are truncated at the top of each graph.

temporal discretizations for inference, which deviate from the training discretization, with high accuracy. In conclusion, our SMDP-SDE model with bidirectional training yields the best performance for a wide range of hyperparameter settings. In appendix F, we additionally evaluate the effects of a logarithmic time discretization and a physics-conditioned score field.

## 4.2 BUOYANCY-DRIVEN FLOW WITH OBSTACLES

Next, we test our methodology on a more challenging problem. For this purpose we consider simulations of buoyancy-driven flow within a fixed domain $\Omega \subset [0,1] \times [0,1]$ and randomly placed

obstacles. We make use of semi-Lagrangian advection for the velocity and MacCormack advection for the hot marker density. The temperature dynamics of the marker field are modeled with a Boussinesq approximation. Each simulation runs from time $t = 0.0$ to $t = 0.65$ with a step size of $\Delta t = 0.01$. The inflow at $(0.2, 0.5)$ is active until $t = 0.2$. Our objective is to employ SMDP-ODE and SMDP-SDE to obtain trajectories that reconstruct a plausible flow given an end state of the marker density and velocity fields at time $t = 0.65$.

Our training data set consists of 250 simulations with corresponding trajectories. For the data generation, we make use of the differentiable *phiflow* solver (Holl et al., 2020a). We place spheres and boxes with varying sizes at different positions within the simulation domain that do not overlap with the marker inflow. For each simulation, we place one to two objects of each category. The testing set comprises 5 simulations. In contrast to the previous task, we generate the training data set without any noise, but add a Gaussian random noise with standard deviation $\sigma = \sqrt{\Delta t}$ to each simulation state of the training trajectories.

**Training and Results**   We employ a neural network architecture based on dilated convolutions (Stachenfeld et al., 2021) for $s_\theta(\mathbf{x}, t)$, see appendix E. The physics operator $\mathcal{P}(\mathbf{x})$ is implemented by using a negative step size $-\Delta t$ for time integration. The training of SMDP-ODE is analogous to the previous experiments on heat equation in section 4.1. However, we do not unroll the entire trajectory and apply a sliding window with size

| Configuration | MSE↓ | | LPIPS↓ $[10^{-3}]$ | |
|---|---|---|---|---|
| | ODE | SDE | ODE | SDE |
| SMDP | **0.05** | 0.78 | **5.82** | 147.17 |
| Rollout noise | 0.12 | 1.03 | 9.10 | 145.72 |
| Physics cond. score | 0.15 | 1.25 | 5.94 | 360.54 |
| Bidir. training | 0.4 | 0.61 | 24.19 | 134.21 |
| Decoupled score&phys. | 0.07 | 0.72 | 7.78 | 67.23 |
| Fully learned | 0.28 | - | 12.54 | - |
| Physics only | 0.5 | - | 346.31 | - |

**Table 2:** Evaluation of variants for the buoyancy obstacle case in terms of reconstruction MSE and LPIPS of the marker field.

20. For this case, we compare our method to a fully learned baseline method. A quantitative evaluation in terms of reconstruction MSE and LPIPS metrics (Zhang et al., 2018) is given in table 2. It becomes apparent that our ODE method clearly outperform the learned baseline. Interestingly, the SDE variant performs less well for this test case. This behavior can be explained by the highly nonlinear system dynamics, and the comparatively approximate reverse simulator which yields the substantial errors for the *Physics only* version in table 2. These errors causes the score network to inadvertently learn significant corrections of the physics operator, which deteriorates the quality of the score field. Nonetheless, as qualitatively shown in figure 6, both variants are able to accurately recover the initial states despite the complex motion of the fluid around the obstacles.

**Algorithmic variants**   We evaluate several altered configurations of our SMDP method to further illustrate its behavior. We consider adding *noise during rollouts*, i.e. adding a noise term $\sigma \cdot z$ for $z \sim \mathcal{N}(0, I)$ to the state $\mathbf{x}$ after applying the score and physics updates during training. Additionally, we experiment with a *physics conditioned score*, i.e. we extend the input dimension of the score function to accept concatenated inputs of the form $s_\theta([\mathbf{x}, \mathcal{P}(\mathbf{x})], t)$. We also evaluate the effects of the bidirectional training. Finally, we evaluate a version that *decouples score and physics*, i.e. we do not evaluate the score and physics update on the same input $\mathbf{x}$, but instead we first apply the physics update $\mathcal{P}(\mathbf{x})$, and evaluate the score function afterwards. Overall, the error measurements in table 2 justify our choices for the baseline SMDP algorithm.

## 4.3   ISOTROPIC TURBULENCE

As third example, we consider a problem where the physics operator is unknown, i.e. we approximate both $\mathcal{P}$ and the score $\nabla_\mathbf{x} \log p_t(\mathbf{x})$ by neural networks. We consider the problem of learning the time evolution of isotropic, forced turbulence as determined by the 2D Navier-Stokes equations with a viscosity of $\nu = 10^{-5}$ (Li et al., 2020). The training data set consists of vorticity fields from 1000 simulation trajectories from $t = 0$ until $T = 10$ with $\Delta t = 1$ and a spatial resolution of $64 \times 64$. Our objective is to predict a trajectory $\hat{\mathbf{x}}_{0:10}$ that reconstructs the true trajectory $\mathbf{x}_{0:10}$ given an end state $\mathbf{x}_{10}$ of the solution, whereas in the original paper, the objective was to learn an operator mapping predicting the vorticity at a later point in time.

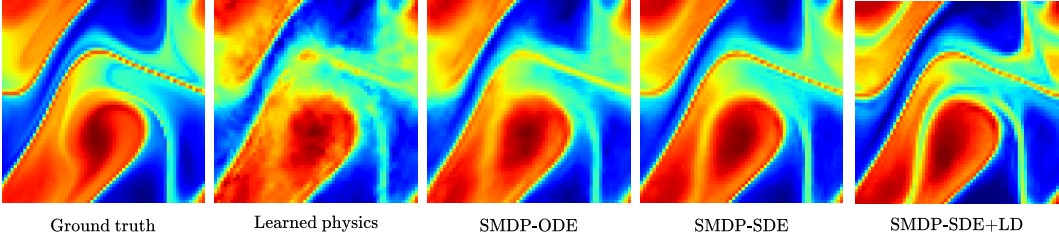

| Ground truth | Learned physics | SMDP-ODE | SMDP-SDE | SMDP-SDE+LD |

**Figure 7:** Turbulence case. Comparison of reconstructed trajectories at $t = 9$.

**Methods** We represent the physics operator $\mathcal{P}(\mathbf{x})$ by a FNO neural network and the score by the ResNet-based architecture from section 4.1. As a first step of our method, we optimize the physics model $\mathcal{P}$ using our SMDP training setup from the previous two experiments, but fix the score to $s_\theta(\mathbf{x}, t) \equiv 0$. We denote this method by *Learned physics*. As a second step, we train SMDP-ODE with the *decoupled score and physics variant* by optimizing the time-dependent score $s_\theta(\mathbf{x}, t)$ while freezing the physics model $\mathcal{P}$. This approach guarantees that any time-independent physics are captured by $\mathcal{P}$ and $s_\theta(\mathbf{x}, t)$ can focus on learning small improvements to $\mathcal{P}$ as well as respond to possibly time-dependent data biases. As this test employs two trained networks for both components of the SDE, we compare to *DiffFlow* (Zhang & Chen, 2021) as an additional baseline.

**Evaluation** We give an evaluation of the improvements of SMDP over the learned variants in table 3. Compared to the Learned physics variant, our methods improve the mean squared error (MSE) between the ground truth trajectories and the reconstructed trajectories slightly, while there is an substantial decrease in the spectral error. This can be seen qualitatively in figure 7. In this scenario DiffFlow has severe difficulties to learn state updates and score field, resulting in large differences in terms of MSE. As before, the SMDP-SDE method performs best in terms of spectral error at the expense of a slightly increased MSE.

| Method | MSE↓ $[10^{-2}]$ | Spectral Error↓ |
|---|---|---|
| SMDP-ODE | **14.3** | 0.26 |
| SMDP-SDE | 15.3 | **0.17** |
| DiffFlow | 221.2 | 0.58 |
| Learned physics | 16.3 | 0.30 |

**Table 3:** Evaluation of the turbulence case.

**Outlook: Refinement with Langevin dynamics** Since the score $\nabla_\mathbf{x} \log p_t(\mathbf{x})$ represents a data gradient, we can use gradient-based optimization methods to find local optima of the data distribution $p_t(\mathbf{x})$ that are close to $\mathbf{x}$. Inspired by stochastic gradient Langevin dynamics (Welling & Teh, 2011), we consider the iteration rule $\mathbf{x}_t^{i+1} = \mathbf{x}_t^i + \epsilon \cdot \nabla_\mathbf{x} \log p_t(\mathbf{x}_t^i) + \sqrt{2\epsilon} z_t$, for $\epsilon = 2 \times 10^{-5}$, where $z_t \sim \mathcal{N}(0, I)$ (details in appendix H.1). Denoted by *SMDP-SDE+LD* in figure 7, this method manages to extract even finer details from the reverse-time SDE solution. As such it provides an interesting starting point for a further refinement of the SMDP results.

## 5 CONCLUSION

We presented SMDP, a derivative of score matching in the context of physical simulations and differentiable physics. We demonstrated its competitiveness against different baseline methods and in challenging inverse physics problems. We demonstrated the versatility of SMDP with two variants: while the *neural ODE* variant focuses on high MSE accuracies, the *neural SDE* variant allows for sampling the posterior and yields an improved coverage of the target data manifold. Despite the promising initial results, our work gives rise to many interesting questions. In particular, the time discretizations is a crucial issue, as for training data generation and differentiable solvers, we would favor larger step size and less evaluations due to computational constraints, while for conventional diffusion models, a large number of smaller time steps often yields substantial improvements in terms of quality. Determining a good balance between accurate solutions with few time steps and diverse solutions with many time steps will remain an important direction for future research in this area.

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

# APPENDIX

## A   ADDITIONAL DETAILS OF TRAINING METHODOLOGY

Below we summarize the problem formulation from the main paper, and provide details about the training procedure together with further information how the solution of the probability flow ODE solution relates to solutions of the SDE.

**Problem setting**   We consider the time evolution of the physical system modeled by the stochastic differential equation

$$dx = \mathcal{P}(x)dt + g(t)dw, \tag{10}$$

with a drift $\mathcal{P} : \mathbb{R}^d \to \mathbb{R}^d$ and diffusion $g : [0, T] \to \mathbb{R}^+$, which transforms the marginal distribution $p_0$ of initial states at time $0$ to the marginal distribution $p_T$ of end states at time $T$. Moreover, we assume that we have sampled $N$ trajectories of length $M$ from the above SDE with a fixed time discretization $0 \le t_0 < t_1 < ... < t_M \le T$ for the interval $[0, T]$ and collected them in a training data set $\{(x_{t_i,n})_{i=0,...,M}\}_{n=0,...,N}$.

We are interested in training a neural network $s_\theta(x, t)$ parameterized by $\theta$ to approximate the score $\nabla_x \log p_t(x)$, i.e. minimize the score matching objective

$$\mathcal{J}_{\mathrm{SM}}(\theta; \lambda(\cdot)) := \frac{1}{2} \int_0^T \mathbb{E}_{x \sim p_t} \left[ \lambda(t) \, ||s_\theta(x, t) - \nabla_x \log p_t(x)||_2^2 \right] dt, \tag{11}$$

where $\lambda : [0, T] \mapsto \mathbb{R}^+$ is a weighting function. In the case of densoing score matching, where the underlying SDE is $dx = f(x, t)dt + g(t)dw$ for affine functions $f(\cdot)$ and $g(\cdot)$, the score can be learned by minimizing the denoising score matching objective using transition kernels, which enables an efficient training of diffusion models, see Song et al. (2021c) for a reference.

In our case $\mathcal{P}$ is an arbitrary function describing the dynamics of the physical system, and hence we can not rely on an analytical expression for the transition kernel.

**Training via Continuous Normalizing Flows**   Score-based diffusion models can be transformed into continuous normalizing flows (Chen et al., 2018, CNFs), which allows for a tractable computation of the likelihood. We can train the corresponding CNF given by

$$dx = \left[ \mathcal{P}(x) - \frac{1}{2} g^2(t) s_\theta(x, t) \right] dt \tag{12}$$

using maximum likelihood training, i.e. maximizing

$$\mathbb{E}_{x_0 \sim p_0}[\log p_0^{\mathrm{ODE}}(x_0)] \tag{13}$$

$$\text{s.t.} \quad x_T = x_0 + \int_0^T \mathcal{P}(x_t) - \frac{1}{2} g^2(t) s_\theta(x_t, t) dt, \tag{14}$$

where $p_0^{\mathrm{ODE}}$ is the distribution obtained by sampling $x_T \sim p_T$ and simulating $x_0$ using ODE equation (12). The log-likelihood can be computed using the instantaneous change of variables formula (Chen et al., 2018) and by using the fact that $p_T$ is approximately Gaussian. For denoising score matching, maximum likelihood training of the corresponding CNF is usually not done, because it requires running an ODE solver for every optimization step and the training with the denoising score matching objective is more efficient (Huang et al., 2021; Song et al., 2021a). It was shown by Song et al. (2021a) that there is connection between the Kullback-Leibler divergence and the score matching objective. In particular

$$\mathrm{KL}(p_0 \,||\, p_\theta^{\mathrm{SDE}}) \le \mathcal{J}_{\mathrm{SM}}(\theta; g(\cdot)^2) + \mathrm{KL}(p_T \,||\, \pi) \tag{15}$$

for a prior distribution $\pi$ and $p_\theta^{\mathrm{SDE}}$ defined by sampling $x_T \sim \pi$ and solving the reverse-time SDE using the score approximation $s_\theta(x, t)$.

**Chaining CNFs**   For SDE equation (10), the distribution $p_T$ describes the perturbed end states and an exact likelihood computation is no longer possible. Equivalent to maximizing the likelihood of the CNF, we can minimize the Kullback-Leibler divergence $\text{KL}(p_0 \,||\, p_0^{\text{ODE}})$.

Since in our problem setting, we do not require $p_0$ or $p_T$ to be a simple distribution for which we can evaluate the likelihood, we can think of the CNF from time $0$ to time $T$ as multiple smaller CNFs chained together, e.g. we consider a chain of CNFs for the time discretizations $t_1$ to $t_0$, $t_2$ to $t_1$ and so on. Each of those CNFs transforms the marginal probabilities $p_{t_i}$ to $p_{t_{i+1}}$ via the probability flow ODE (12) by minimizing $\text{KL}(p_{t_i} \,||\, p_{t_i}^{\text{ODE},t_{i+1}})$ during training, where $p_{t_i}^{\text{ODE},t_{i+1}}$ is defined by sampling $\mathbf{x}_{t_{i+1}} \sim p_{t_{i+1}}$ and simulating the probability flow ODE from time $t_{i+1}$ until $t_i$.

For example, we can sample $\mathbf{x_T} \sim p_T$ and generate data from $p_0$ by recursively using the CNFs to generate $\mathbf{x}_{t_i}$ given $\mathbf{x}_{t_{i+1}}$ until we reach $\mathbf{x}_0$. Since CNFs are bijective, this also works in the reverse direction, i.e. we can sample $\mathbf{x}_0 \sim p_0$ and simulate $\mathbf{x}_0$ from $p_0$ with the same method.

In the following, we will derive our method using the regular, increasing flow of time, i.e. $0 \to T$. The direction $T \to 0$ follows analogously by replacing the SDE (10) with the corresponding reverse-time SDE (Anderson, 1982). Then, the new objective obtained by chaining the CNFs becomes the minimization of

$$\sum_{i=0}^{M-1} \text{KL}(p_{t_{i+1}} \,||\, p_{t_{i+1}}^{\text{ODE},t_i}), \tag{16}$$

where $p_{t_{i+1}}^{\text{ODE},t_i}$ is now defined for the time increasing direction, i.e. sampling $\mathbf{x}_{t_i} \sim p_{t_i}$ and simulating $\mathbf{x}_{t_{i+1}}$ via the probability flow ODE. If $s_\theta(\mathbf{x}, t) \equiv \nabla_{\mathbf{x}} \log p_t(\mathbf{x})$, then by the theory of probability flow ODEs (Song et al., 2021c), the above objective will become $0$, since the marginal probabilities $p_t$ of the SDE will coincide with the marginal likelihoods of the probability flow CNF at every time $t$. In this case, then we also obtain $\text{KL}(p_T \,||\, p_T^{\text{ODE}}) = 0$, i.e. when sampling $\mathbf{x}_{t_0} \sim p_0$ and solving the probability flow ODE, we obtain exactly the distribution $p_T$.

**Additional assumptions**   In the following, we make the same additionally assumptions as Song et al. (2021a), Appendix A. Specifically, we require that

(i) $\exists C_p > 0 \; \forall \mathbf{x} \in \mathbb{R}^d \; t \in [0, T] : ||\nabla_{\mathbf{x}} \log p_t(\mathbf{x})||_2 \le C_p (1 + ||\mathbf{x}||_2)$

(ii) $\exists C_s > 0 \; \forall \theta \; \forall \mathbf{x} \in \mathbb{R}^d \; t \in [0, T] : ||s_\theta(\mathbf{x}, t)||_2 \le C_s (1 + ||\mathbf{x}||_2)$

(iii) $\exists C_\mathcal{P} > 0 \; \forall \mathbf{x} \in \mathbb{R}^d : ||\mathcal{P}(\mathbf{x})||_2 \le C_\mathcal{P} (1 + ||\mathbf{x}||_2)$

(iv) $\forall t \in [0, T] \; \exists k > 0 : p_t(\mathbf{x}) \in O(e^{-||\mathbf{x}||_2^k})$ as $||\mathbf{x}||_2 \to \infty$

**Minimizing the Kullback-Leibler divergence**   The $i$-th summand of equation (16) can be simplified to

$$\text{KL}(p_{t_{i+1}} \,||\, p_{t_{i+1}}^{\text{ODE},t_i}) = \mathbb{E}_{\mathbf{x}_{t_{i+1}} \sim p_{t_{i+1}}} \left[ \log \left( \frac{p_{t_{i+1}}(\mathbf{x}_{t_{i+1}})}{p_{t_{i+1}}^{\text{ODE},t_i}(\mathbf{x}_{t_{i+1}})} \right) \right] \tag{17}$$

$$= \mathbb{E}_{\mathbf{x}_{t_i} \sim p_{t_i}} \mathbb{E}_{\mathbf{x}_{t_{i+1}} \sim p_{t_{i+1}} | \mathbf{x}_{t_i}} \left[ \log \left( \frac{p_{t_{i+1}}(\mathbf{x}_{t_{i+1}})}{p_{t_{i+1}}^{\text{ODE},t_i}(\mathbf{x}_{t_{i+1}})} \right) \right] \tag{18}$$

$$= -\mathbb{E}_{\mathbf{x}_{t_i} \sim p_{t_i}} \mathbb{E}_{\mathbf{x}_{t_{i+1}} \sim p_{t_{i+1}} | \mathbf{x}_{t_i}} \left[ \log \left( p_{t_{i+1}}^{\text{ODE},t_i}(\mathbf{x}_{t_{i+1}}) \right) \right] + C, \tag{19}$$

where $C$ is a constant independent of $\theta$. Thus, we can maximize the expectation:

$$\mathbb{E}_{\mathbf{x}_{t_i} \sim p_{t_i}} \mathbb{E}_{\mathbf{x}_{t_{i+1}} \sim p_{t_{i+1}} | \mathbf{x}_{t_i}} \left[ \log \left( p_{t_{i+1}}^{\text{ODE},t_i}(\mathbf{x}_{t_{i+1}}) \right) \right] \tag{20}$$

**Locality of CNFs and estimating $p_{t_{i+1}}^{\text{ODE},t_i}(\mathbf{x}_{t_{i+1}})$**   With the law of iterated expectations, we can write the probability $p_{t_{i+1}}^{\text{ODE},t_i}(\mathbf{x}_{t_{i+1}})$ in equation (20) as

$$p_{t_{i+1}}^{\text{ODE},t_i}(\mathbf{x}_{t_{i+1}}) = \mathbb{E}_{\hat{\mathbf{x}}_{t_i} \sim p_{t_i}} \left[ p_{t_{i+1}}^{\text{ODE},t_i}(\mathbf{x}_{t_{i+1}} | \hat{\mathbf{x}}_{t_i}) \right]. \tag{21}$$

We show in the following that we can approximate the expectation in equation (21) with only negligible error $\kappa > 0$ on a different distribution $\tilde{p}_{t_i}(\mathbf{x}_{t_{i+1}})$ that depends on $\mathbf{x}_{t_{i+1}}$ and $\kappa$. Importantly, the support of $\tilde{p}_{t_i}(\mathbf{x}_{t_{i+1}})$ is a bounded set with $\mathrm{diam}(\mathrm{supp}(\tilde{p}_{t_i}(\mathbf{x}_{t_{i+1}}))) \to 0$ as $t_{i+1} - t_i \to 0$ for all $\mathbf{x}_{t_{i+1}} \in \mathbb{R}^d$. Intuitively, the score $\nabla_{\mathbf{x}} \log p_t(\mathbf{x})$ is determined by the (local) probabilities $p_t$ in a small environment around $\mathbf{x}$ instead of the global properties of $p_t$. This insight is an important part of our methodology and loss estimation.

To prove this, given $\mathbf{x}_{t_{i+1}}$, we first define a set of points $\mathbf{x}$ for which the density $p_{t_{i+1}}^{\mathrm{ODE},t_i}(\mathbf{x}_{t_{i+1}}|\mathbf{x})$ is greater than a chosen $\kappa > 0$, i.e. we define

$$S^\kappa(\mathbf{x}_{t_{i+1}}) = \left\{ \mathbf{x} \in \mathbb{R}^d \,|\, p_{t_{i+1}}^{\mathrm{ODE},t_i}(\mathbf{x}_{t_{i+1}}|\mathbf{x}) \geq \kappa \right\}. \tag{22}$$

Then, for this set, we consider the indicator function $I_{\mathbf{x}_{t_{i+1}}}^\kappa(\cdot)$, which is 1 for elements in $S^\kappa(\mathbf{x}_{t_{i+1}})$ and 0 otherwise. Then, we can rewrite equation (21) as

$$\mathbb{E}_{\hat{\mathbf{x}}_{t_i} \sim p_{t_i}} \left[ p_{t_{i+1}}^{\mathrm{ODE},t_i}(\mathbf{x}_{t_{i+1}}|\hat{\mathbf{x}}_{t_i}) \right] \tag{23}$$

$$= \mathbb{E}_{\hat{\mathbf{x}}_{t_i} \sim p_{t_i}} \left[ I_{\mathbf{x}_{t_{i+1}}}^\kappa(\hat{\mathbf{x}}_{t_i}) p_{t_{i+1}}^{\mathrm{ODE},t_i}(\mathbf{x}_{t_{i+1}}|\hat{\mathbf{x}}_{t_i}) + (1 - I_{\mathbf{x}_{t_{i+1}}}^\kappa(\hat{\mathbf{x}}_{t_i})) p_{t_{i+1}}^{\mathrm{ODE},t_i}(\mathbf{x}_{t_{i+1}}|\hat{\mathbf{x}}_{t_i}) \right]. \tag{24}$$

By the definition of the set $S^\kappa(\mathbf{x}_{t_{i+1}})$, we can derive the following bounds

$$\mathbb{E}_{\hat{\mathbf{x}}_{t_i} \sim p_{t_i}} \left[ (1 - I_{\mathbf{x}_{t_{i+1}}}^\kappa(\hat{\mathbf{x}}_{t_i})) p_{t_{i+1}}^{\mathrm{ODE},t_i}(\mathbf{x}_{t_{i+1}}|\hat{\mathbf{x}}_{t_i}) \right] \leq \kappa \tag{25}$$

and

$$\mathbb{E}_{\hat{\mathbf{x}}_{t_i} \sim p_{t_i}} \left[ I_{\mathbf{x}_{t_{i+1}}}^\kappa(\hat{\mathbf{x}}_{t_i}) p_{t_{i+1}}^{\mathrm{ODE},t_i}(\mathbf{x}_{t_{i+1}}|\hat{\mathbf{x}}_{t_i}) + (1 - I_{\mathbf{x}_{t_{i+1}}}^\kappa(\hat{\mathbf{x}}_{t_i})) p_{t_{i+1}}^{\mathrm{ODE},t_i}(\mathbf{x}_{t_{i+1}}|\hat{\mathbf{x}}_{t_i}) \right] \tag{26}$$

$$\geq \mathbb{E}_{\hat{\mathbf{x}}_{t_i} \sim p_{t_i}} \left[ I_{\mathbf{x}_{t_{i+1}}}^\kappa(\hat{\mathbf{x}}_{t_i}) p_{t_{i+1}}^{\mathrm{ODE},t_i}(\mathbf{x}_{t_{i+1}}|\hat{\mathbf{x}}_{t_i}) \right]. \tag{27}$$

$$\geq p_{t_{i+1}}^{\mathrm{ODE},t_i}(\mathbf{x}_{t_{i+1}}) - \kappa \tag{28}$$

.

For the next part, we need to make an approximation for $p_{t_{i+1}}^{\mathrm{ODE},t_i}(\cdot|\mathbf{x}_{t_i})$.

**Approximating $p_{t_{i+1}}^{\mathrm{ODE},t_i}(\cdot|\mathbf{x}_{t_i})$ as a Gaussian** Since the CNF is bijective, choosing $p_{t_{i+1}}^{\mathrm{ODE},t_i}(\cdot|\mathbf{x}_{t_i})$ as a Dirac delta function with spike located at $\mu_{\mathrm{ODE}}(\mathbf{x}_{t_i})$ is the correct choice. Here, $\mu_{\mathrm{ODE}}(\mathbf{x}_{t_i})$ is defined as the solution of the probability flow ODE equation (12) for $\mathbf{x}_{t_i}$ integrated from time $t_i$ to $t_{i+1}$. However, we are assuming that we are limited by machine precision and inexact ODE solvers to compute $\mu_{\mathrm{ODE}}(\mathbf{x})$ anyway. Therefore, we make the assumption that given $\mathbf{x}_{t_i}$, solving the probability flow ODE equation (12) until $t_{i+1}$ will give a solution $\mathbf{x}_{t_{i+1}}^{\mathrm{ODE}}$ that is approximately Gaussian with mean $\mu_{\mathrm{ODE}}(\mathbf{x}_{t_i}) = \mathbf{x}_{t_i} + (t_{i+1} - t_i)(\mathcal{P}(\mathbf{x}_{t_i}) - \frac{1}{2} g_{t_i}^2 s_\theta(\mathbf{x}_{t_i}, t_i))$ and standard deviation $\sigma_{\mathrm{ODE}} = \epsilon(t_{i+1} - t_i)$, for an arbitrary, but small $\epsilon > 0$. This approximation makes use of the explicit Euler method and thus also relies on the time step $t_{i+1} - t_i$ being sufficiently small to ensure stability of the time integration.

Given the above approximation with Euler steps, we can derive the following bound on the distance between $\mathbf{x}$ and $\mu_{\mathrm{ODE}}(\mathbf{x})$ using assumptions (i), (ii) and (iii)

$$||\hat{\mathbf{x}}_{t_i} - \mu_{\mathrm{ODE}}(\hat{\mathbf{x}}_{t_i})||_2 = ||\hat{\mathbf{x}}_{t_i} - \hat{\mathbf{x}}_{t_i} - (t_{i+1} - t_i)(\mathcal{P}(\hat{\mathbf{x}}_{t_i}) + \tfrac{1}{2} g_{t_i} s_\theta(\hat{\mathbf{x}}_{t_i}, t))||_2 \tag{29}$$

$$\leq (t_{i+1} - t_i)(||\mathcal{P}(\hat{\mathbf{x}}_{t_i})||_2 + \tfrac{1}{2} g_{t_i}^2 ||s_\theta(\hat{\mathbf{x}}_{t_i}, t))||_2) \tag{30}$$

$$\leq (t_{i+1} - t_i) \left[ (1 + C_\mathcal{P})||\hat{\mathbf{x}}_{t_i}||_2 + \tfrac{1}{2} g_{t_i}^2 (1 + C_s)||\hat{\mathbf{x}}_{t_i}||_2 \right] \tag{31}$$

$$\leq (t_{i+1} - t_i) C_{\mathrm{ODE}} ||\hat{\mathbf{x}}_{t_i}||_2. \tag{32}$$

Morover, the approximation using Gaussians gives us the following equivalence for the term $p_{t_{i+1}^{\mathrm{ODE}},t_i}(\mathbf{x}_{t_{i+1}}|\hat{\mathbf{x}}_{t_i})$, which we have used to define the set $S^\kappa(\mathbf{x}_{t_{i+1}})$ in equation (22)

$$p_{t_{i+1}^{\mathrm{ODE}},t_i}(\mathbf{x}_{t_{i+1}}|\hat{\mathbf{x}}_{t_i}) \geq \kappa \tag{33}$$

$$\iff \frac{1}{\sqrt{(2\pi)^d |\Sigma|}} \exp\left(\mathbf{x}_{t_{i+1}} - \mu_{\mathrm{ODE}}(\hat{\mathbf{x}}_{t_i})\right) \Sigma^{-1} (\mathbf{x}_{t_{i+1}} - \mu_{\mathrm{ODE}}(\hat{\mathbf{x}}_{t_i}))^T \geq \kappa, \tag{34}$$

where $\Sigma = \sigma_{\text{ODE}}^2 I$. The above is then equivalent to

$$||x_{t_{i+1}} - \mu_{\text{ODE}}(\hat{\mathbf{x}}_{t_i})||_2^2 \leq -\log\left(\kappa\sqrt{(2\pi)^d \sigma_{\text{ODE}}^{2d}}\right)\sigma_{\text{ODE}}^2 \tag{35}$$

Note that $\sigma_{\text{ODE}}$ depends on $\epsilon$ and $(t_{i+1} - t_i)$.

We can now finally define the distribution $\tilde{p}_{t_i}(\mathbf{x}_{t_{i+1}})$. For this, we define the set

$$\tilde{S}^\kappa(\mathbf{x}_{t_{i+1}}) := \bigcup_{\mathbf{x} \in S^\kappa(\mathbf{x}_{t_{i+1}})} \left\{\tilde{\mathbf{x}} \in \mathbb{R}^d \,|\, ||\tilde{\mathbf{x}} - \mathbf{x}||_2 \leq (t_{i+1} - t_i)C_{\text{ODE}}||\tilde{\mathbf{x}}||_2\right\}, \tag{36}$$

Now, by combining equation (35) and equation (32), we obtain $\text{diam}(\tilde{S}^\kappa(\mathbf{x}_{t_{i+1}})) \to 0$ as $t_{i+1} - t_i \to 0$. For the indicator function $\tilde{I}_{\mathbf{x}_{t_{i+1}}}^\kappa(\cdot)$ on $\tilde{S}^\kappa(\mathbf{x}_{t_{i+1}})$, we get the bound

$$I_{\mathbf{x}_{t_{i+1}}}^\kappa(\mathbf{x}) \leq \tilde{I}_{\mathbf{x}_{t_{i+1}}}^\kappa(\mathbf{x}) \quad \forall \mathbf{x} \in \mathbb{R}^d \tag{37}$$

and therefore also

$$\mathbb{E}_{\hat{\mathbf{x}}_{t_i} \sim p_{t_i}}\left[(1 - \tilde{I}_{\mathbf{x}_{t_{i+1}}}^\kappa(\hat{\mathbf{x}}_{t_i}))p_{t_{i+1}}^{\text{ODE},t_i}(\mathbf{x}_{t_{i+1}}|\hat{\mathbf{x}}_{t_i})\right] \leq \kappa. \tag{38}$$

Given $\tilde{S}^\kappa(\mathbf{x}_{t_{i+1}})$, we define the distribution $\tilde{p}_{t_i}(\mathbf{x}_{t_{i+1}})$ based on $p_{t_i}$ but with support restricted to $\tilde{S}^\kappa(\mathbf{x}_{t_{i+1}})$. Then, we get the following equality

$$\mathbb{E}_{\hat{\mathbf{x}}_{t_i} \sim p_{t_i}}\left[\tilde{I}_{\mathbf{x}_{t_{i+1}}}^\kappa(\hat{\mathbf{x}}_{t_i})p_{t_{i+1}}^{\text{ODE},t_i}(\mathbf{x}_{t_{i+1}}|\hat{\mathbf{x}}_{t_i})\right] = \mathbb{E}_{\hat{\mathbf{x}}_{t_i} \sim \tilde{p}_{t_i}(\mathbf{x}_{t_{i+1}})}\left[C_{\mathbf{x}_{t+1}} p_{t_{i+1}}^{\text{ODE},t_i}(\mathbf{x}_{t_{i+1}}|\hat{\mathbf{x}}_{t_i})\right], \tag{39}$$

where $C_{\mathbf{x}_{t+1}}$ is a normalizing constant depending on $\kappa$, $t_{i+1} - t_i$ and $\mathbf{x}_{t+1}$.

**Maximizing the variational lower bound** From Jensen's inequality, equation (39) and equation (38), we obtain a lower bound on equation (20)

$$\mathbb{E}_{\mathbf{x}_{t_i} \sim p_{t_i}}\mathbb{E}_{\mathbf{x}_{t_{i+1}} \sim p_{t_{i+1}}|\mathbf{x}_{t_i}}\left[\log\left(p_{t_{i+1}}^{\text{ODE},t_i}(\mathbf{x}_{t_{i+1}})\right)\right] \tag{40}$$

$$\geq \mathbb{E}_{\mathbf{x}_{t_i} \sim p_{t_i}}\mathbb{E}_{\mathbf{x}_{t_{i+1}} \sim p_{t_{i+1}}|\mathbf{x}_{t_i}}\left[\log\left(\mathbb{E}_{\hat{\mathbf{x}}_{t_i} \sim \tilde{p}_{t_i}(\mathbf{x}_{t_{i+1}})}\left[C_{\mathbf{x}_{t+1}} p_{t_{i+1}}^{\text{ODE},t_i}(\mathbf{x}_{t_{i+1}}|\hat{\mathbf{x}}_{t_i})\right]\right)\right] \tag{41}$$

$$\geq \mathbb{E}_{\mathbf{x}_{t_i} \sim p_{t_i}}\mathbb{E}_{\mathbf{x}_{t_{i+1}} \sim p_{t_{i+1}}|\mathbf{x}_{t_i}}\mathbb{E}_{\hat{\mathbf{x}}_{t_i} \sim \tilde{p}_{t_i}(\mathbf{x}_{t_{i+1}})}\left[\log\left(C_{\mathbf{x}_{t+1}} p_{t_{i+1}}^{\text{ODE},t_i}(\mathbf{x}_{t_{i+1}}|\hat{\mathbf{x}}_{t_i})\right)\right] \tag{42}$$

$$= \mathbb{E}_{\mathbf{x}_{t_i} \sim p_{t_i}}\mathbb{E}_{\mathbf{x}_{t_{i+1}} \sim p_{t_{i+1}}|\mathbf{x}_{t_i}}\mathbb{E}_{\hat{\mathbf{x}}_{t_i} \sim \tilde{p}_{t_i}(\mathbf{x}_{t_{i+1}})}\left[\log\left(p_{t_{i+1}}^{\text{ODE},t_i}(\mathbf{x}_{t_{i+1}}|\hat{\mathbf{x}}_{t_i})\right) + \log(C_{\mathbf{x}_{t+1}})\right], \tag{43}$$

which is the same as maximizing

$$\mathbb{E}_{\mathbf{x}_{t_i} \sim p_{t_i}}\mathbb{E}_{\mathbf{x}_{t_{i+1}} \sim p_{t_{i+1}}|\mathbf{x}_{t_i}}\mathbb{E}_{\hat{\mathbf{x}}_{t_i} \sim \tilde{p}_{t_i}(\mathbf{x}_{t_{i+1}})}\left[\log\left(p_{t_{i+1}}^{\text{ODE},t_i}(\mathbf{x}_{t_{i+1}}|\hat{\mathbf{x}}_{t_i})\right)\right] \tag{44}$$

Thus, instead of the original objective equation (20), we instead maximize the lower bound from equation (44).

**Deriving the L2 loss** Since $p_{t_{i+1}}^{\text{ODE},t_i}(\cdot|\hat{\mathbf{x}}_{t_i})$ is the density of a Gaussian, the objective equation (44) is equivalent to minimizing

$$\mathbb{E}_{\mathbf{x}_{t_i} \sim p_{t_i}}\mathbb{E}_{\mathbf{x}_{t_{i+1}} \sim p_{t_{i+1}}|\mathbf{x}_{t_i}}\mathbb{E}_{\hat{\mathbf{x}}_{t_i} \sim \tilde{p}_{t_i}(\mathbf{x}_{t_{i+1}})}\left[||\mathbf{x}_{t_{i+1}} - \mu_{\text{ODE}}(\hat{\mathbf{x}}_{t_i})||_2^2/\sigma_{\text{ODE}}^2\right] \tag{45}$$

$$\propto \mathbb{E}_{\mathbf{x}_{t_i} \sim p_{t_i}}\mathbb{E}_{\mathbf{x}_{t_{i+1}} \sim p_{t_{i+1}}|\mathbf{x}_{t_i}}\mathbb{E}_{\hat{\mathbf{x}}_{t_i} \sim \tilde{p}_{t_i}(\mathbf{x}_{t_{i+1}})}\left[||\mathbf{x}_{t_{i+1}} - \mu_{\text{ODE}}(\hat{\mathbf{x}}_{t_i})||_2^2\right], \tag{46}$$

where

$$\mu_{\text{ODE}}(\hat{\mathbf{x}}_{t_i}) = \hat{\mathbf{x}}_{t_i} + (t_{i+1} - t_i)\left[\mathcal{P}(\hat{\mathbf{x}}_{t_i}) - \frac{1}{2}g_{t_i}^2 s_\theta(\hat{\mathbf{x}}_{t_i}, t_i)\right]. \tag{47}$$

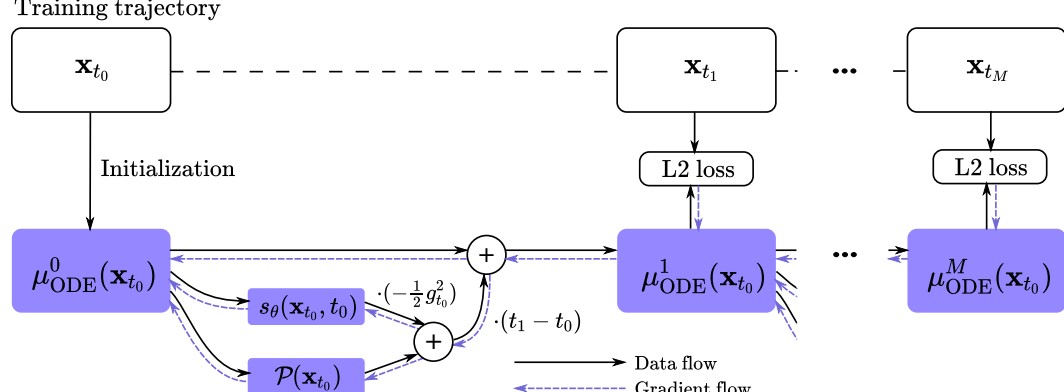

**Figure 8:** Training overview for the trajectory $(\mathbf{x}_{t_0}, \mathbf{x}_{t_1}, ..., \mathbf{x}_{t_M})$. Gradients are backpropagated over multiple time steps via automatic differentiation. This requires that the physics operator $\mathcal{P}$ is differentiable. Incoming gradients at $s_\theta(\mathbf{x}_{t_i}, t_i)$ are used to obtain gradients for $\theta$, which are summed over all steps $i$. The network weights $\theta$ are then updated based on the optimizer, e.g. stochastic gradient descent or Adam.

**Estimating the loss**  The training data set $\{(\mathbf{x}_{t_i,n})_{i=0,...,M}\}_{n=0,...,N}$ is sampled directly from the SDE of the physical system equation (10), so the empirical distribution $p_{t_i}^{\mathrm{emp}}$ induced by the training data set at time discretization $t_i$ is close to the marginal distribution $p_{t_i}$ for $0 \leq i \leq M$ and sufficiently large $N$. We make use of this fact to approximate the sampling of $\mathbf{x}_{t_i} \sim p_{t_i}$ and $\mathbf{x}_{t_{i+1}} \sim p_{t_{i+1}}|\mathbf{x}_{t_i}$ in equation (46), by drawing a data sample $\mathbf{x}_{t_i}$ at time $t_i$ and its next successor $\mathbf{x}_{t_{i+1}}$ on the trajectory at time $t_{i+1}$ from the training data set.

We approximate sampling from the distribution $\tilde{p}_{t_i}(\mathbf{x}_{t_{i+1}})$ by reusing the sampled $\mathbf{x}_{t_i}$. Intuitively, for the CNF from $t_i$ to $t_{i+1}$, our optimization fits a deterministic (bijective) process defined by the probability flow to the non-deterministic SDE trajectories, which start at the same point.

**Extension to multiple time steps**  We jointly train multiple time steps. First, we sample a trajectory $(\mathbf{x}_{t_i}, ..., \mathbf{x}_{t_j})$ with $0 \leq i < j \leq M$ from the training data set. Then, the loss becomes

$$\sum_{k=i+1}^{j} ||\mathbf{x}_{t_k} - \mu_{\mathrm{ODE}}^k(\mathbf{x}_{t_i})||_2^2, \tag{48}$$

where $\mu_{\mathrm{ODE}}^k(\mathbf{x}_{t_i})$ is the discretized trajectory from the probability flow ODE, i.e.

$$\mu_{\mathrm{ODE}}^i(\mathbf{x}_{t_i}) = \mathbf{x}_{t_i} \tag{49}$$

$$\mu_{\mathrm{ODE}}^k(\mathbf{x}_{t_i}) = \mu_{\mathrm{ODE}}^{k-1}(\mathbf{x}_{t_i}) + (t_k - t_{k-1})\left[\mathcal{P}(\mu_{\mathrm{ODE}}^{k-1}(\mathbf{x}_{t_i})) - \frac{1}{2}g_{t_k}^2 s_\theta(\mu_{\mathrm{ODE}}^{k-1}(\mathbf{x}_{t_i}), t_k)\right]. \tag{50}$$

For multiple steps, instead of reusing $\mathbf{x}_{t_i}$ as a sample from $\tilde{p}_{t_i}(\mathbf{x}_{t_{i+1}})$, we use the previous ODE solution $\mu_{\mathrm{ODE}}^k(\mathbf{x}_{t_i})$. An overview of the training for multiple steps is shown in Figure 8.

**Bidirectional training**  Since CNFs are bidirectional and our loss formulation does not make any specific assumptions about $p_{t_i}$ and $p_{t_{i+1}}$, we can consider training the reverse time direction. In this case, the SDE describing the physical system becomes the reverse-time SDE. Analogous to the increasing time direction, we sample the trajectory $(\mathbf{x}_{t'_i}, ..., \mathbf{x}_{t'_j})$, but the time direction is reversed, i.e. $t'_i > t'_j$ and $t_0 = t'_M$ and $t_M = t'_0$. We train the combined objective by sampling a trajectory from the training data set and alternating between optimizing the time-forward direction equation (48), and the corresponding time-backwards direction loss. Note that training both the directions (i.e. $p_0$ to $p_T$ and vice versa) has been explored in other works as well, for example for training Schrödinger Bridges (Bortoli et al., 2021; Chen et al., 2022), where the discrepancy between the final marginal distribution (i.e. either $p_T^{\mathrm{SDE}}$ or $p_0^{\mathrm{SDE}}$) for the time-forward and respectively time-backward (or reverse-time) SDE is minimized.

**Rollout length**  The training of SMDP requires unrolling algorithm 1 as shown in figure 8. Additionally, we adopt a training method based on sliding windows. For a window size $S$, we consider the points $(\mathbf{x}_{t_M}, \mathbf{x}_{t_{M-1}}, ..., \mathbf{x}_{t_{M-S}})$ from a training trajectory and unroll the SMDP algorithm 1 for $S$ steps. We compute the loss equation (48) and backpropagate gradients through all steps to obtain updates for $\theta$. Then, we move the window by 1, i.e. consider the points $(\mathbf{x}_{t_{M-1}}, \mathbf{x}_{t_{M-2}}, ..., \mathbf{x}_{t_{M-S-1}})$ and compute the updates for $\theta$. We repeat the above $M$ times until we have covered the entire trajectory. If the training data trajectories are short, we use $S = M$, otherwise, we pick a lower window size $S$.

If $S = 1$, then we do not require differentiability of the physics operator $\mathcal{P}$, as SMDP reduces to predicting the next point on the trajectory given the previous point.

Starting with an untrained score network, long rollouts may yield divergent trajectories because of the physics dynamics. Therefore losses may be very high and the training becomes unstable. We therefore typically start training with a short sliding window, e.g. 2. We train for a few epochs and then increase the sliding window size by a constant. We repeat this until we reach a sufficiently high rollout length, which yields stable trajectories for the entire simultion.

We give details of this for each specific experiment either directly in the main text or the accompanying appendix.

**Comparison to DiffFlow**  Zhang & Chen (2021) train DiffFlow by considering forward and backward processes in the context of generative modelling. In their setting, the drift $f(\mathbf{x}, t)$ is also a learnable neural network and they consider $p_0$ to be the data distribution and $p_T$ to resemble a simple noise distribution. Specifically, they implement the forward and backward processes as

$$\mathbf{x}_{i+1} = \mathbf{x}_i + f_i(\mathbf{x}_i)\Delta t_i + g_i \delta_i^F \sqrt{\Delta t_i} \tag{51}$$

$$\mathbf{x}_i = \mathbf{x}_{i+1} - [f_{i+1}(\mathbf{x}_{i+1}) - g_{i+1}^2 s_{i+1}(\mathbf{x}_{i+1})]\Delta t_i + g_{i+1}\delta_{i+1}^B \sqrt{\Delta t_i} \tag{52}$$

for two samples $\delta_i^F, \delta_i^B \sim \mathcal{N}(0, I)$ and time discretization $\{t_i\}_{i=0}^N$ and $\Delta t_i = (t_{i+1} - t_i)$.

Zhang & Chen (2021) directly minimize the KL divergence between the trajectory distribution for the forward and backward process, i.e. they minimize

$$\mathrm{KL}(p_F(\tau)||p_B(\tau)) = \mathbb{E}_{\tau \sim p_F}[\log p_F(\mathbf{x}_0)] + \mathbb{E}_{\tau \sim p_B}[\log p_B(\mathbf{x}_N)] \tag{53}$$

$$+ \sum_{i=1}^{N-1} \mathbb{E}_{\tau \sim p_F} \left[ \log \frac{p_F(\mathbf{x}_i|\mathbf{x}_{i-1})}{p_B(\mathbf{x}_{i-1}|\mathbf{x}_i)} \right] \tag{54}$$

Using the forward and backward process discretizations as well as the fact that $p_B(\mathbf{x}_N)$ is a Gaussian distribution, they are able to derive a loss based on the squared difference between the forward and backward process as well as an additional likelihood term for $-\log p_B(\mathbf{x}_N)$, see equation (15) in their paper.

Our method on the other hand directly minimizes the KL divergence between marginal distribution $p_t$ and the ones produced by deterministic probability flow ODE $p_t^{\mathrm{ODE}}$. Thus, our loss likewise minimizes the difference between a forward and backward trajectory. Moreover, in our case, $p_T$ is not constrained to a simple noise distribution.

# B  ADDITIONAL EXPERIMENT: 1D PROCESS

We discuss an additional experiment, where we compare SMDP with Implicit Score Matching (ISM) (Hyvärinen, 2005). For this task, we consider the SDE given by

$$dx = -\left[\lambda_1 \cdot \text{sign}(x)x^2\right]dt + \lambda_2 dw. \tag{55}$$

The corresponding reverse-time SDE is given by

$$dx = -\left[\lambda_1 \cdot \text{sign}(x)x^2 - \lambda_2^2 \cdot \nabla_x \log p_t(x)\right]dt + \lambda_2 dw. \tag{56}$$

Throughout this experiment, $p_0$ is a categorical distribution, where we draw either $1$ or $-1$ with the same probability. In figure 9, we show trajectories from this SDE simulated with the Euler-Maruyama method. Trajectories either start at $1$ or $-1$ and approach $0$ as $t$ increases. Given the trajectory value at $t = 10$, it is no longer possible to infer the origin of the trajectory at $t = 0$.

We employ a neural network $s_\theta(x, t)$ parameterized by $\theta$ to approximate the score via ISM and compare it to SMDP. In both cases, the neural network is a simple multilayer perceptron with tanh activations and 5 hidden layers with 20 neurons for the first four hidden layers and 10 neurons for the last hidden layer.

Our training data set consists of 250 simulated trajectories from $t = 0$ until $t = 10$ and $\Delta t = 0.02$. Therefore each training trajectory has length $M = 500$.

**Implicit Score Matching**   Implicit Score Matching (Hyvärinen, 2005) is a score matching method that leverages the fact that for a random vector $\mathbf{x} \in \mathbb{R}^d$ with probability density function $p$, minimizing the score matching objective

$$J(\theta) := \frac{1}{2}\mathbb{E}_{\mathbf{x}\sim p}\left[||s_\theta(\mathbf{x}) - \nabla_\mathbf{x} \log p(x)||_2^2\right] \tag{57}$$

is equivalent to minimizing the following objective

$$J'(\theta) := \mathbb{E}_{\mathbf{x}\sim p}\left[\sum_{i=1}^d \frac{\partial s_\theta(\mathbf{x})_i}{\partial \mathbf{x}_i} + \frac{1}{2}s_\theta(\mathbf{x})_i^2\right]. \tag{58}$$

Note that for ISM, there is no explicit time dimension, so we are absorbing the time dimension into $\mathbf{x}$, i.e. for the trajectory $(x_1, x_2, ..., x_M)$ sampled at time $(t_1, t_2, ..., t_M)$, we concatenate value and time with $\mathbf{x}^{(i)} := (x_i, t_i) \in \mathbb{R}^2$. We collect $\mathbf{x}_i$ for $i = 1, ..., M$ from all trajectories in a new training data set. When training $s_\theta(\mathbf{x})$, we therefore lose the information, to which trajectory a value-time pair originally belonged. We obtain the time-dependent score $\nabla_{\mathbf{x}_1} \log p_{\mathbf{x}_2}(\mathbf{x}_1)$ from the first coordinate of the output of $s_\theta(\mathbf{x})$, i.e. $s_\theta(\mathbf{x})_1$.

We compute the partial derivative $\partial s_\theta(\mathbf{x})_i/\partial \mathbf{x}_i$ using reverse-mode automatic differentiation in JAX (jax.jacrev). We train $s_\theta(\mathbf{x})$ with the Adam optimizer for 15.000 epochs with learning rate $10^{-3}$, which we decrease by a factor of 0.1 every 5.000 epochs.

**SMDP Training**   The training of SMDP follows the experiments in section 4 with slight problem-specific modifications. Since the length of the trajectories is very long ($M = 500$), we subsample the trajectories, and keep every 5th point in time. We train with a sliding window of size 4, which we increase every 500 epochs by 2 until we reach an window size of 40 with the Adam optimizer and learning rate $10^{-5}$. Then, we finetune the network and train on the complete trajectories for an additional 500 epochs and sliding window size 70 with learning rate $10^{-6}$.

**Comparison**   We show a direct comparison of the learned score, the reverse-time SDE trajectories and the probability flow trajectories between ISM and SMDP in figure 10. The learned score of ISM and SMDP in figure 10a and figure 10b is similar until $t = 2$. Then, the score for both networks becomes positive for points marginally above 0, i.e. the score pushes points up, leading them to 1 at $t = 0$. Analogously, points marginally below 0 are pushed down. For SMDP, the absolute value of the score in this region is considerably higher than ISM, thus having a more significant shearing effect.

This affects the reverse-time SDE trajectories. Because of the stronger shearing, SMDP trajectories result in states of either $1$ or $-1$, see figure 10d. On the other hand, for ISM in figure 10c, where the shearing is less pronounced, many trajectories end up in-between $1$ and $-1$. These states are not valid samples from the posterior distribution of the solution from equation (55).

Interestingly, the probability flow solutions for both SMDP and ISM have comparable quality, cf. figure 10e and figure 10f. In both cases, trajectories that start above $0$ at $t = 10$ end close to $1$ at $t = 0$, and vice versa.

Overall, this case illustrates that the learned score of SMDP is more accurate than the one of ISM, because of its stronger shearing. This results in a better quality of the reverse-time SDE trajectories. SMDP performs better here, because it is directly trained with an integration of the physics dynamics, whereas ISM is purely data-driven and does not incorporate the physics model with its temporal evolutions at training time.

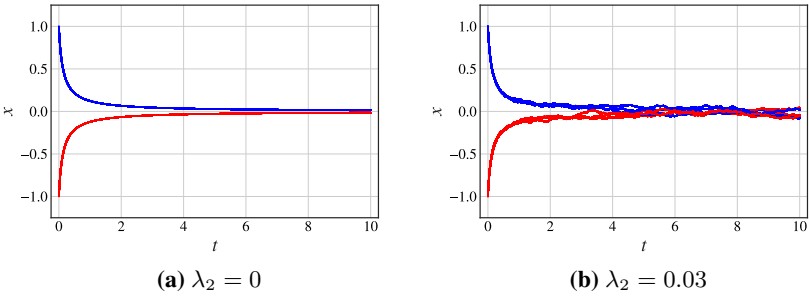

**Figure 9:** Trajectories from SDE equation (55) with $\lambda_2 = 0$ (a) and $\lambda_2 = 0.03$ (b).

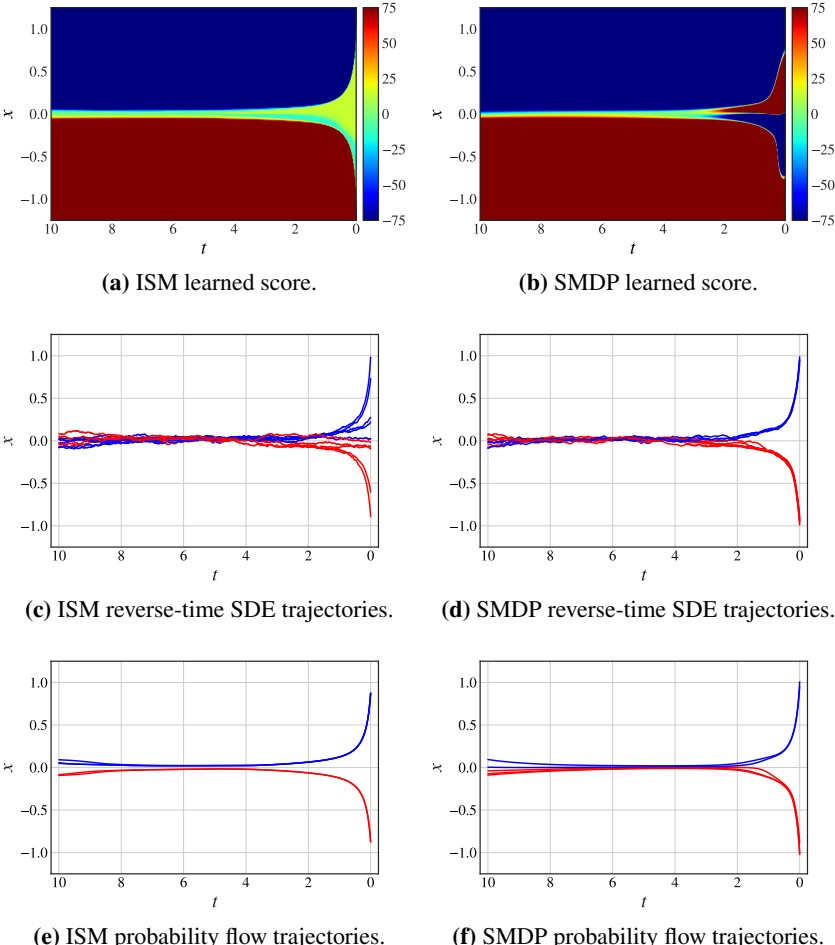

**Figure 10:** Comparison of Implicit Score Matching (ISM, left) and Score Matching via Differentiable Physics (SMDP, right). Colormap in (a) and (b) truncated to [-75, 75].

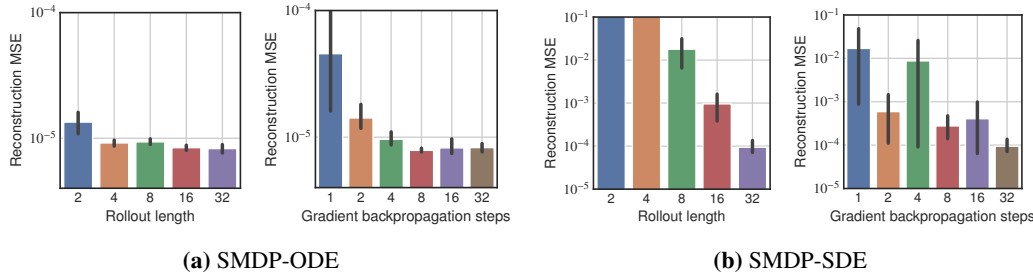

(a) SMDP-ODE          (b) SMDP-SDE

**Figure 11:** Heat equation reconstruction MSE for different sliding window sizes $S$ (Rollout length) and number of gradient backpropagation steps for SMDP-ODE (a) and SMDP-SDE (b).

## C  ABLATION FOR GRADIENT BACKPROPAGATION AND ROLLOUT LENGTH

**Gradient backpropagation steps** It is possible to unroll for $S$ steps, but stop the gradient backpropagation after $G < S$ steps. For example, when unrolling the algorithm for $S = 32$ steps, we can unroll the first $G = 8$ steps, save the intermediate results, then backpropagate the gradient to optimize $\theta$. Then, we continue unrolling with the intermediate results as a new initialization, again stopping and updating $\theta$ after $G$ steps. For this example, we can repeat this $S/G = 4$ times in total until we obtain combined rollout of $S$ steps.

An advantage of stopping the gradient after $G$ steps is that it reduces memory requirements, as we do not need to store all intermediate results and therefore this allows for training with large window sizes.

For all other experiments in the paper, we have set $G = S$ always.

**Effect on reconstruction MSE** We evaluate the effect of changing the window size $S$ and the number of steps until we stop the gradient backpropagation using the setup of the heat equation experiment in section 4.1. In figure 11, we show the results on the reconstruction MSE for SMDP-ODE and SMDP-SDE. We keep the time discretization fixed, i.e. a full simulation trajectory consists of 32 steps from $t = 0$ until $t = 0.2$, however we vary the window size $S$. In this case, we always backpropagate gradients through all unrolling steps. In the second case, where we change the number of gradient backpropagation steps, we keep the window size $S$ fixed at 32, which is the entire simulation trajectory. For both SMDP-ODE and SMDP-SDE, our evaluation shows that both longer rollouts and more gradient backpropagation steps improve the reconstruction MSE. The improvements are more significant for SMDP-SDE.

## D 1D HEAT EQUATION: EVALUATION OF POSTERIOR

To demonstrate the diversity and high quality of the reverse-time SDE trajectories we provide an additional evaluation of the posterior for a one-dimensional heat equation case. We compare the SDE solutions with samples obtained from filtering a large data set. To simplify the comparison, we consider 1D processes based on the 2D Gaussian random fields from section 4.1. Each process has a length of 100 and corresponds to a Gaussian random field of size $1 \times 100$. Analogously to section 4.1, we use the heat equation to simulate the states forward in time from $t = 0.0$ until $t = 0.2$. Figure 12 shows some examples of the 1D processes we consider here.

**Training of SMDP** The network $s_\theta(\mathbf{x}, t)$ representing the score is trained as described for the heat equation experiment, section 4.1, using the same ResNet architecture with padding in $y$-direction removed. We consider a time discretization with $\Delta t = 0.01$. We begin training with an initial rollout of 6 steps and increase the rollout length every 5 epochs by 2 until we reach 14 rollout steps. For every epoch, we train on 20 randomly generated 1D processes. We use the Adam optimizer with learning rate $10^{-4}$.

In the following, we randomly generate a 1D process $P$. We describe how we generate samples for $t = 0.0$ conditioned on the simulation end state of $P$ at $t = 0.2$.

**Reverse-SDE posterior** We initialize the state based on $P$ at $t = 0.2$. Then, we simulate the reverse-time SDE with the learned score $s_\theta(\mathbf{x}, t)$ via Euler-Maruyama steps. A visualization of 100 samples is shown in figure 14.

**Empirical distribution** We sample $10^6$ processes and form pairs of initial state $t = 0$ and end states $t = 0.2$. We filter the 100 process end states that are closest to the end state of P in terms of the L2 distance. As solutions from the empirical sampling, we consider the 100 corresponding initial states as shown in figure 13. This empirical distribution makes a qualitative comparison possible, as shown in figures 14 and 13. They indicate that the reverse-time SDE solutions are diverse while matching P very well. Simulating the obtained solutions forward in time gives end state that are in excellent agreement with P.

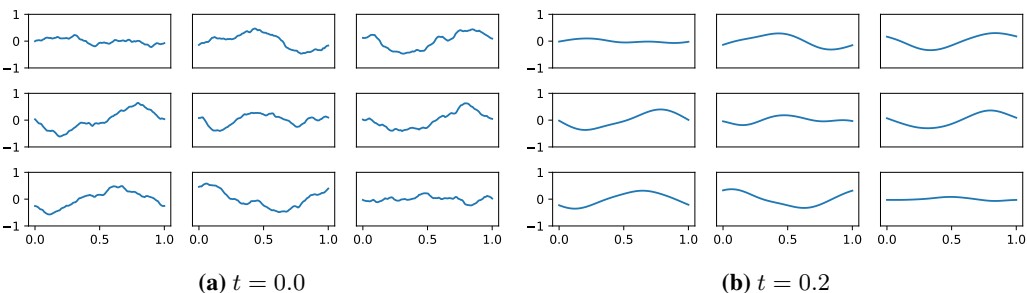

**(a)** $t = 0.0$      **(b)** $t = 0.2$

**Figure 12:** Examples of training data for 1D heat equation with initial states at $t = 0.0$ (a) and end states at $t = 0.2$ (b).

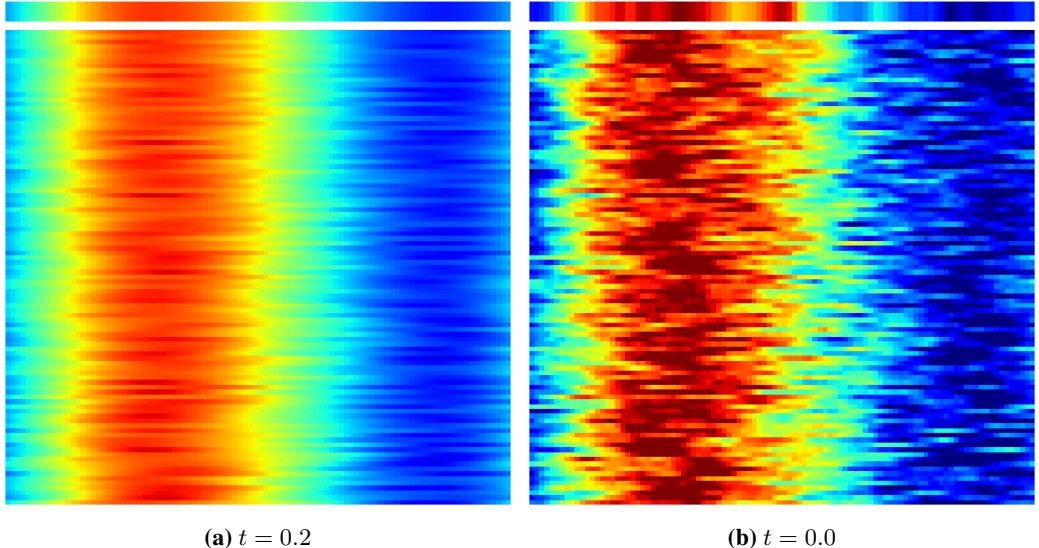

**(a)** $t = 0.2$               **(b)** $t = 0.0$

**Figure 13: Empirical distribution**: we generate $10^6$ Gaussian processes and simulate them forward in time from $t = 0.0$ until $t = 0.2$. We pick one specific process $P$ and sort all other processes in ascending order based on the L2 distance to $P$ at time $t = 0.2$. Then we pick the first 100 and visualize them at time $t = 0.0$ (a) and $t = 0.2$ (b). The top row in both plots shows the process P.

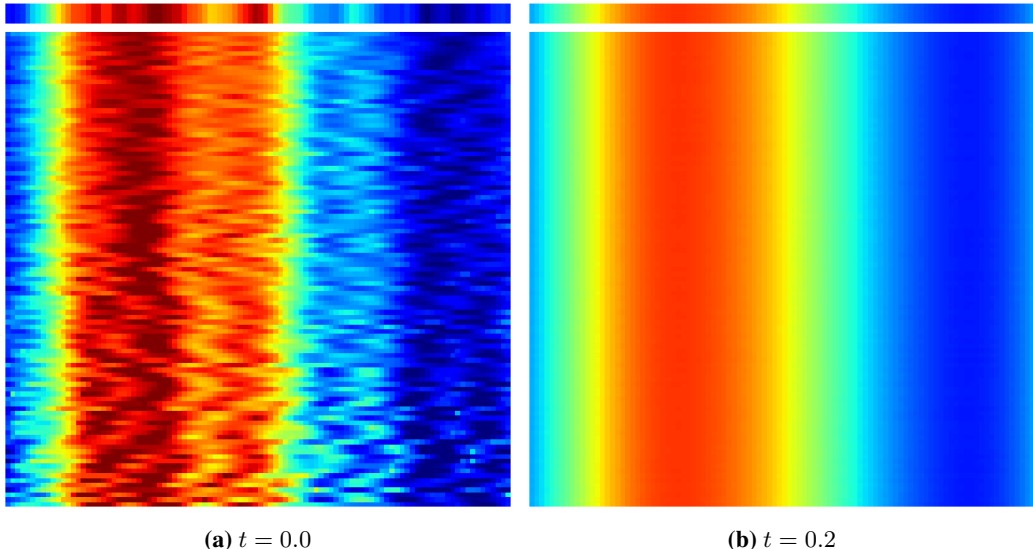

**(a)** $t = 0.0$               **(b)** $t = 0.2$

**Figure 14: Reverse-time SDE**: We pick the process P from figure 13. Then, we simulate 100 trajectories from the reverse-time SDE with learned score and $P$ at $t = 0.2$ as initialization. We sort the states at $t = 0.0$ based on their distance to P and visualize them (a). We then simulate all states from (a) forward in time again until $t = 0.2$, see (b). The forward simulated trajectories almost exactly match P at $t = 0.2$.

# E  ARCHITECTURES

**ResNet**  We employ a simple ResNet-like architecture, which is used in Section 4.1 for the score function $s_\theta(\mathbf{x}, t)$ and the convolutional neural network baseline (ResNet-S and ResNet-P) as well as in Section 4.3 again for the score $s_\theta(\mathbf{x}, t)$.

Since in both experiments, there are periodic boundary conditions, we apply a periodic padding with length 16, i.e. if the underlying 2-dimensional data dimensions are $N \times N$, the dimenions after the periodic padding are $(N+16) \times (N+16)$. We implement the periodic padding by first tiling the input 3 times in $x$- and $y$-direction and then cropping to the correct sizes. The time $t$ is concatenated as an additional constant channel to the 2-dimensional input data when this architecture is used to represent the score $s_\theta(\mathbf{x}, t)$.

The encoder-part of our network begins with a single 2D-convolution encoding layer with 32 filters, kernel size 4 and no activation function. This is followed by 4 consecutive residual blocks, each consisting of 2D-convolution, LeakyReLU, 2D-convolution and Leaky ReLU. All 2D convolutions have 32 filters with kernel size 4 and stride 1. The encoder part ends with a single 2D convolution with 1 filter, kernel size 1 and no activation. Then, in the decoder part, we begin with a transposed 2D convolution, 32 filters, kernel size 4. Afterwards, there are 4 consecutive residual blocks, analogous to the encoder residual blocks, but with the 2D convolution replaced by a transposed 2D convolution. Finally there is a final 2D convolution with 1 filter and kernel size 5. Parameters statistics of this model, as well as the others are given in table 4.

**UNet**  We use the UNet architecture with spatial dropout as given in (Mueller et al., 2022), Appendix A.1. The dropout rate is set to 0.25. We do not include batch normalization and apply the same periodic padding as done for our ResNet architecture.

**FNO**  For all experiments, we consider the FNO-2D architecture introduced in (Li et al., 2020) with $k_{\max,j} = 12$ Fourier modes per channel.

**Dil-ResNet**  The Dil-ResNet architecture is described in (Stachenfeld et al., 2021), Appendix A. Since this architecture represents the score $s_\theta$ in Section 4.2, we concatenate the constant time channel analogously to the ResNet architecture. Additionally, positional information is added to the network input by encoding the $x$-position and $y$-position inside the domain in two separate channels.

| Architecture | Parameters |
|---|---|
| ResNet | 330.754 |
| UNet | 706.035 |
| DilatedConv | 336.915 |
| FNO | 465.377 |

**Table 4:** Summary of architectures.

## F    HEAT EQUATION

### F.1    ADDITIONAL TRAINING DETAILS

**Spectral loss**    We consider a spectral error based on the two-dimensional power spectral density. For two 2d-images, we compute their 2d Fourier transform and compute the radially averaged power spectrum $s_1$ and $s_2$. Then, we define the spectral error as the difference between the log of the spectral densities

$$\mathcal{L}(s_1, s_2) = |\log(s_1) - \log(s_2)|. \tag{59}$$

**SMDP-ODE**    For inference, we consider the linear time discretization $t_n = n\Delta t$ with $\Delta t = 0.2/32$ and $t_{32} = 0.2$. During training, we sample a random time discretization $\tilde{t}_n$ for each batch based on $t_n$ by $\tilde{t}_n \sim \mathcal{U}(t_n - \Delta t/2, t_n + \Delta t/2)$ for $n = 1, ..., 31$ to not overfit on the step size. In the warmup phase of training, we unroll Algorithm 1 for $N = 6, 8, ..., 32$ steps, where we increase $N$ every 2 epochs. We employ Adam to update the weights $\theta$ with learning rate $10^{-4}$. After the warmup is finished, we finetune the network weights for 80 epochs with an initial learning rate of $10^{-4}$ which we reduce by a factor of 0.5 every 20 epochs.

**Diffusion model**    The diffusion model is inspired by (Rissanen et al., 2022). We make use of the same ResNet architecture as SMDP-ODE, however we keep the time discretization $t_n$ fixed. The diffusion model does not include the physics operator $\mathcal{P}$ in the rollout. Therefore, it has to learn the score and physics at the same time. Except for those changes, the training setup is identical to SMDP-ODE.

**Baseline methods**    All other baseline methods are trained for 80 epochs using the Adam optimizer algorithm with an initial learning rate of $10^{-4}$ which is decreased by a factor of 0.5 every 20 epochs. For the training data, we consider solutions to the heat equation consisting of initial state $\mathbf{x}_0$ and end state $\mathbf{x}_T$ that are noise-free and add a Gaussian noise with standard deviation $\sigma = 0.1$ to the network input.

### F.2    LOGARITHMIC TIME DISCRETIZATION

For this experiment, we consider a logarithmic time discretization during training and inference. This discretization is finer around $t = 0$, where most of the small-scale structures are smoothened out. We find that this method also works well, but the linear time discretization performs better for most cases, especially when $\Delta t$ is changed for inference, cf. figure 5 and figure 15.

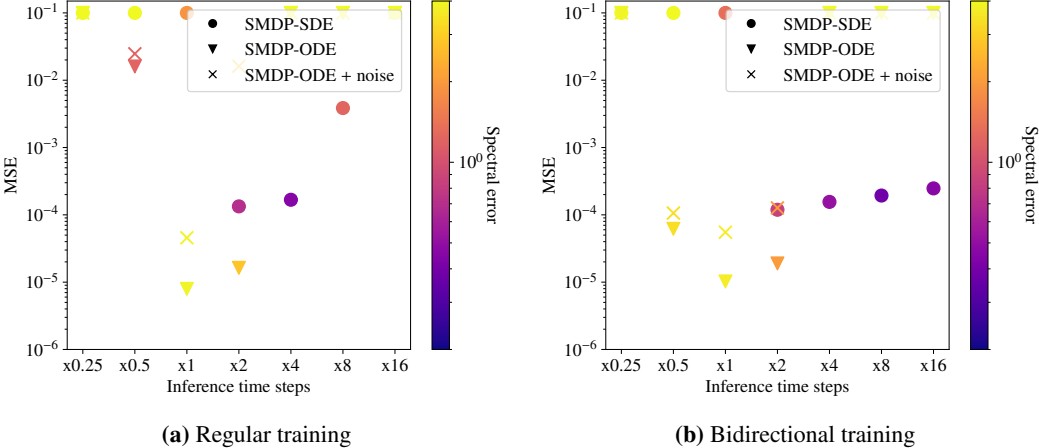

**(a)** Regular training                    **(b)** Bidirectional training

**Figure 15:** Logarithmic time discretization. Reconstruction MSE and spectral errors for varying time steps of regular variant (a) and bidirectional variant (b). Large errors are truncated at the top of each graph.

### F.3 Physics-conditioned score

We extend the definition of the score function, to also include information about the physics at time $t$, i.e. $\mathcal{P}(\mathbf{x}_t)$. Thus, we replace $s_\theta(\mathbf{x}, t)$ in algorithm 1 by $s_\theta([\mathbf{x}, \mathcal{P}(\mathbf{x})], t)$, where we concatenate both inputs. An evaluation is shown in figure 16. Although the stability of SMDP-ODE and SMDP-ODE + noise is greatly increased, we do not obtain results with a low spectral error for SMDP-SDE. Interestingly, the bidirectional training also seems harmful in this case.

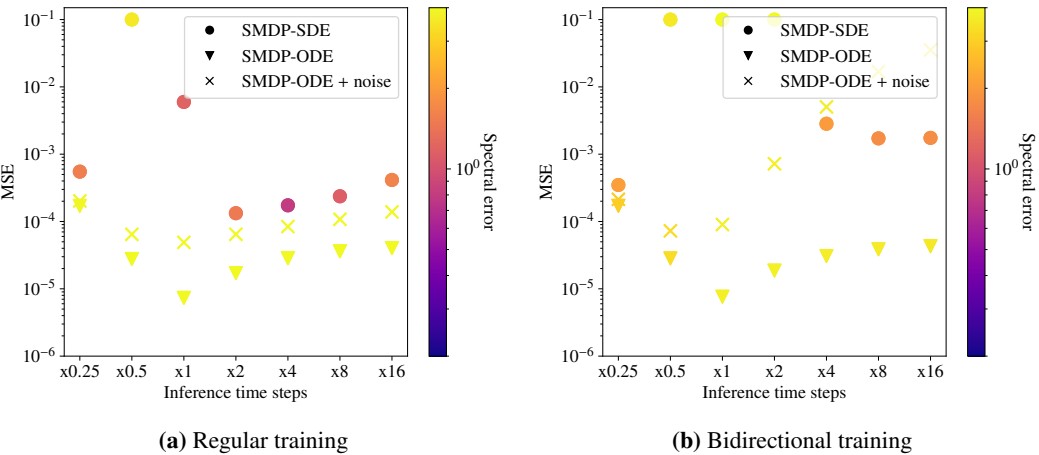

**(a)** Regular training      **(b)** Bidirectional training

**Figure 16:** Physics-conditioned score. Reconstruction MSE and spectral errors for varying time steps of regular variant (a) and bidirectional variant (b). Large errors are truncated at the top of each graph.

### F.4 Higher Training Noise

We increase the diffusion coefficient from $g \equiv 0.1$ to $g \equiv 1.0$ during training, but for inference, we set it back to the lower value. An evaluation is shown in figure 17. Overall, SMDP-SDE produces results with low spectral error and SMDP-ODE has a much lower MSE, similar to the standard case. However, overall, the MSE is much higher, indicating shortcomings of this approach.

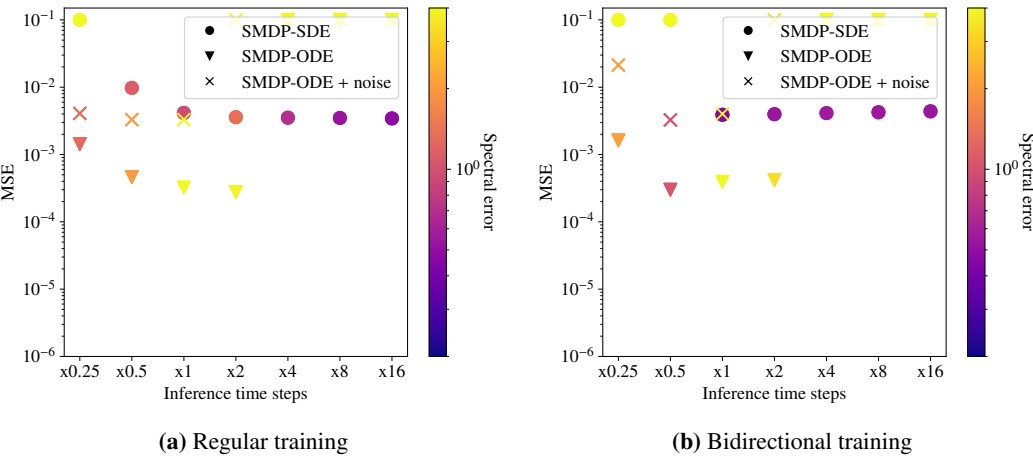

**(a)** Regular training      **(b)** Bidirectional training

**Figure 17:** Higher training noise. Reconstruction MSE and spectral errors for varying time steps of regular variant (a) and bidirectional variant (b). Large errors are truncated at the top of each graph.

### F.5 Additional Results

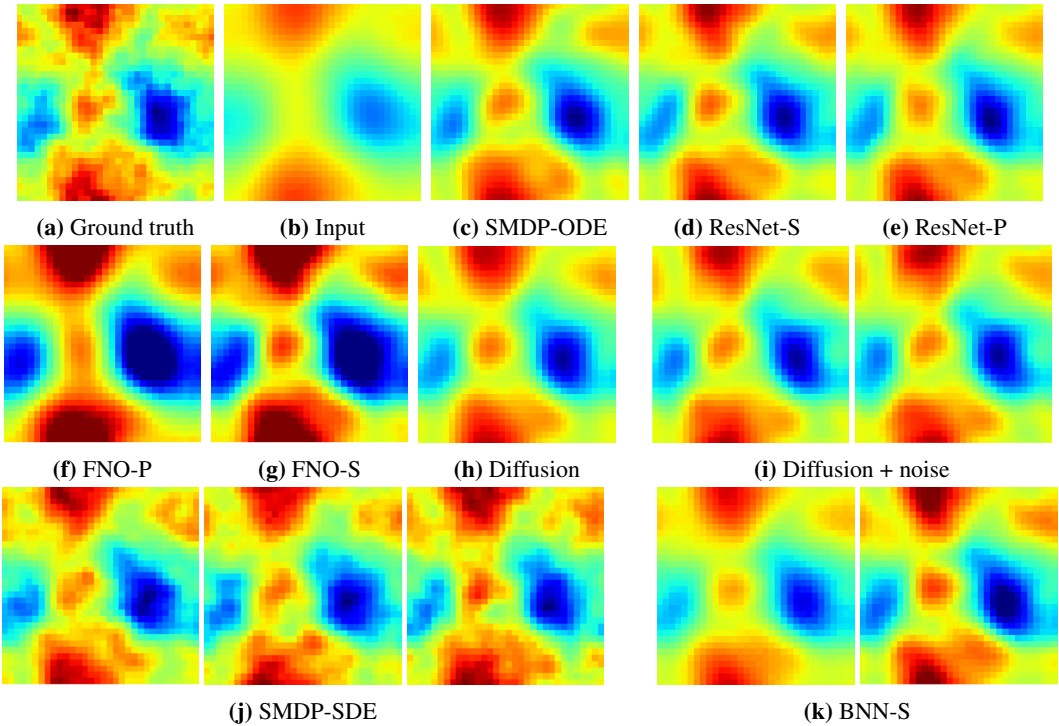

**Figure 18:** Predictions of different methods for the heat equation problem (example 1 of 2).

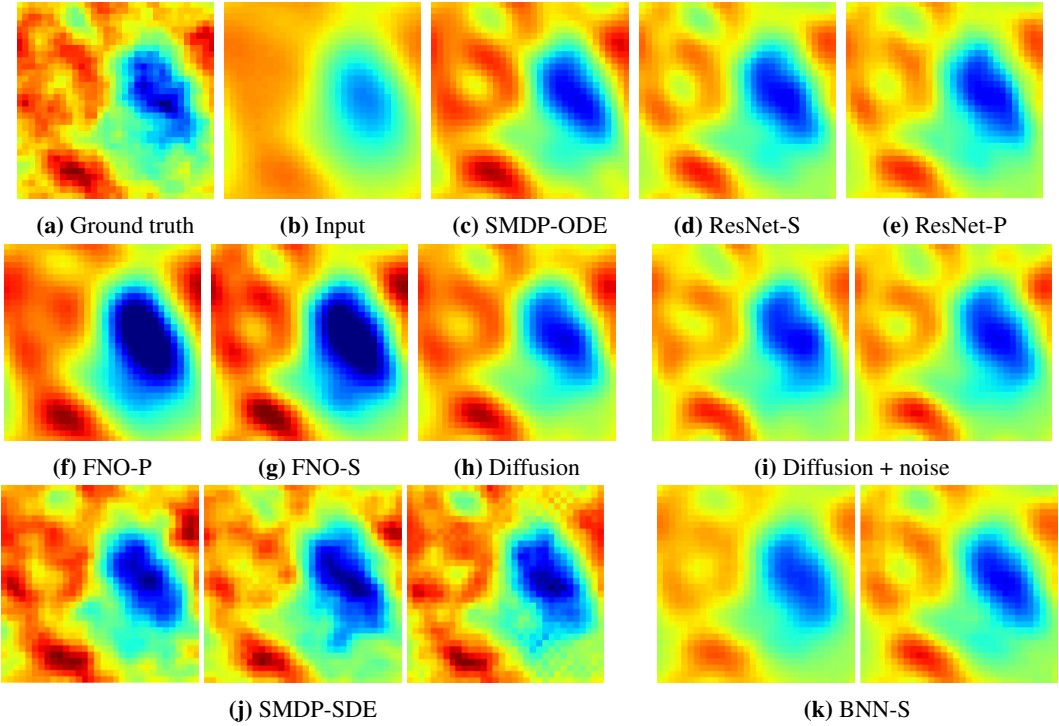

**Figure 19:** Predictions of different methods for the heat equation problem (example 2 of 2). Neither the BNN nor the Diffusion+noise model are able to produce small-scale structures.

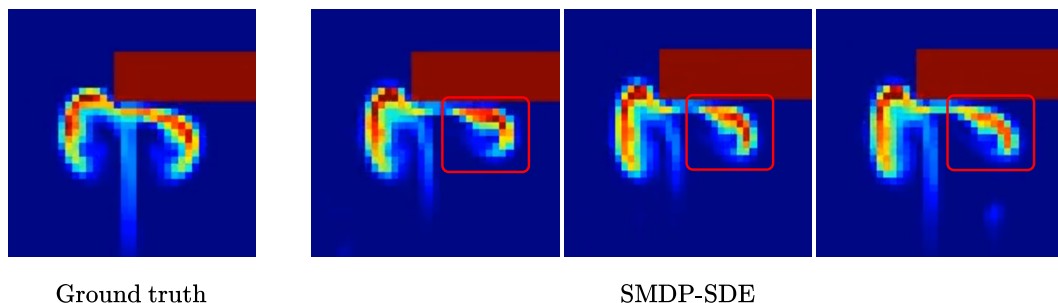

Ground truth                                    SMDP-SDE

**Figure 20:** Comparison of SMDP-SDE predictions and ground truth for buoyancy-driven flow at $t = 0.36$.

## G   BUOYANCY-DRIVEN FLOW WITH OBSTACLES

### G.1   TRAINING

We train all networks with Adam and learning rate $1 \times 10^{-4}$ with batch size 16. We begin with unrolling $N = 2$ steps, which we increase every 30 epochs by 2 until we reach $N = 20$.

### G.2   ADDITIONAL RESULTS

We give more detailed time evolutions of results for the buoyancy-driven flow case in figure 21 and figure 22. These again highlight the difficulties of the physics simulator to recover the initial states by itself. The SMDP variants significantly improve upon this behavior.

In figure 20 we also show an example of the posterior sampling for this case. It becomes apparent that the inferred small-scale structures of the different samples change. However, in contrast to cases like the heat diffusion example, the physics simulation in this scenario leaves only little room for substantial changes of the states.

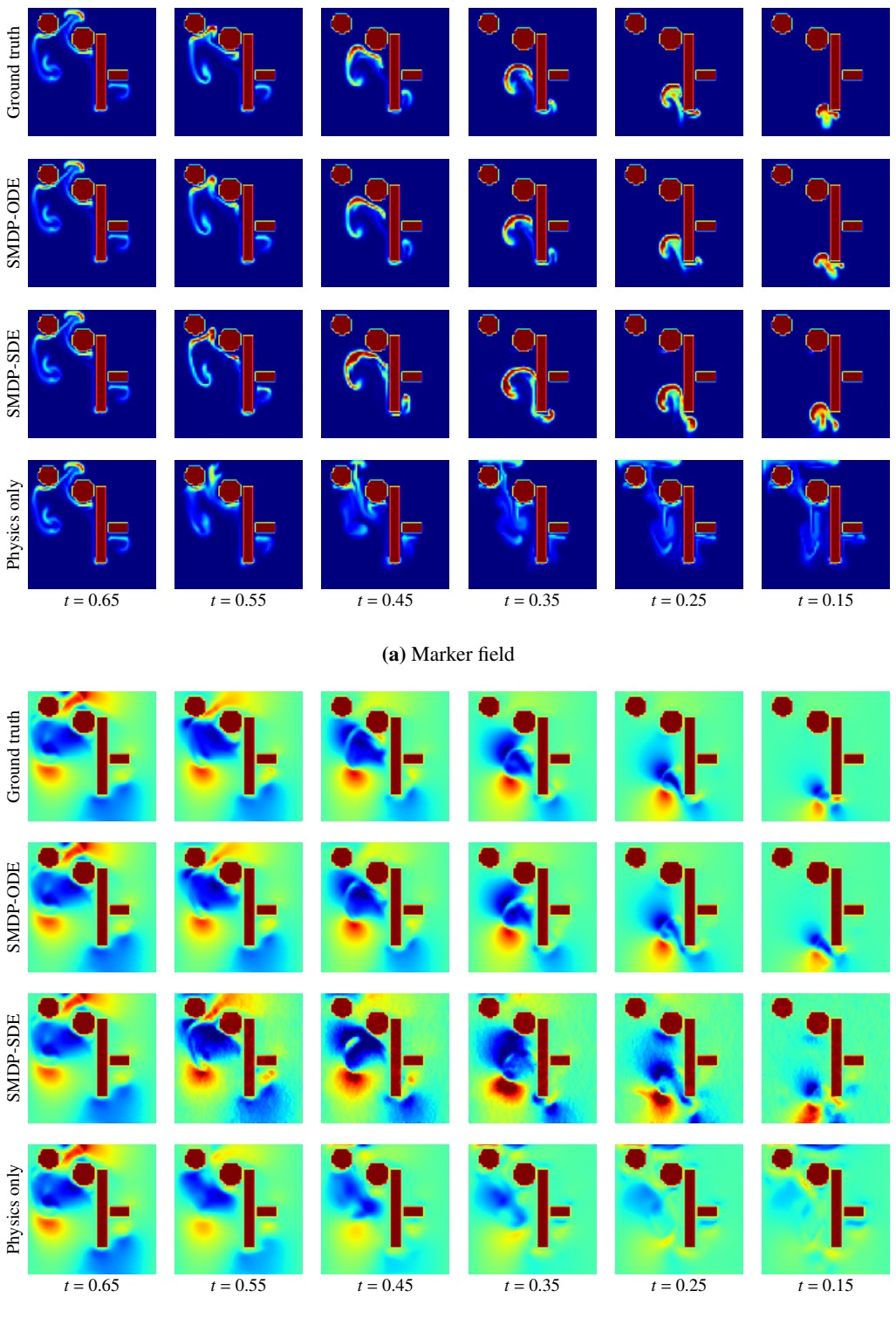

**(a)** Marker field

**(b)** Velocity field ($x$-direction)

**Figure 21:** Predictions for buoyancy-driven flow with obstacles (example 1 of 2).

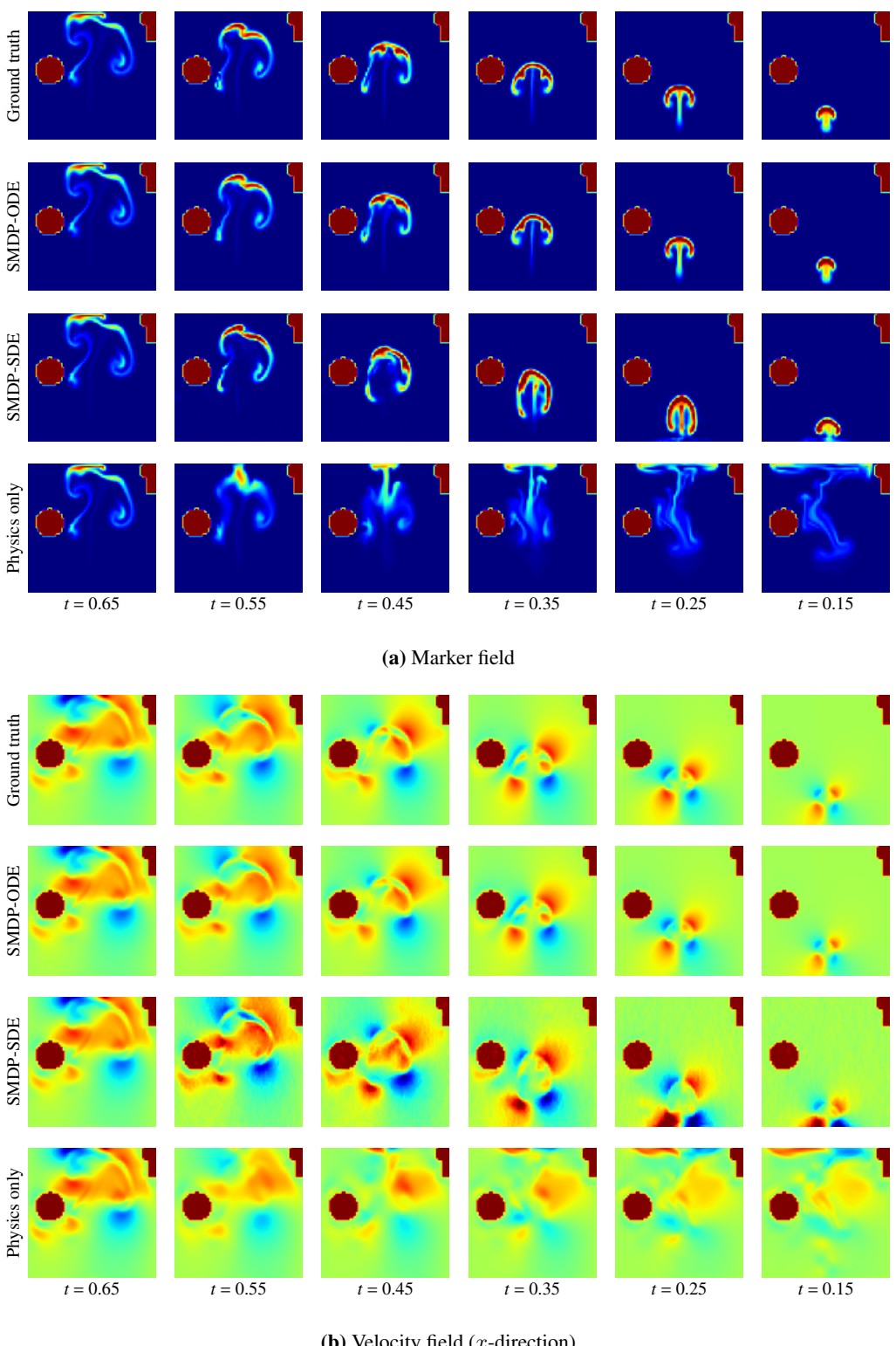

**(a)** Marker field

**(b)** Velocity field ($x$-direction)

**Figure 22:** Predictions for buoyancy-driven flow with obstacles (example 2 of 2).

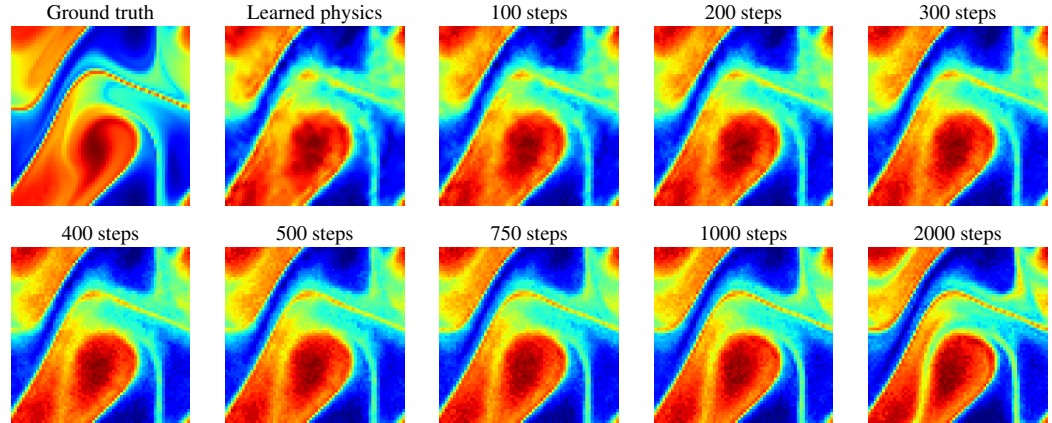

**Figure 23:** Steps of Langevin dynamics for $\epsilon = 2 \times 10^{-5}$.

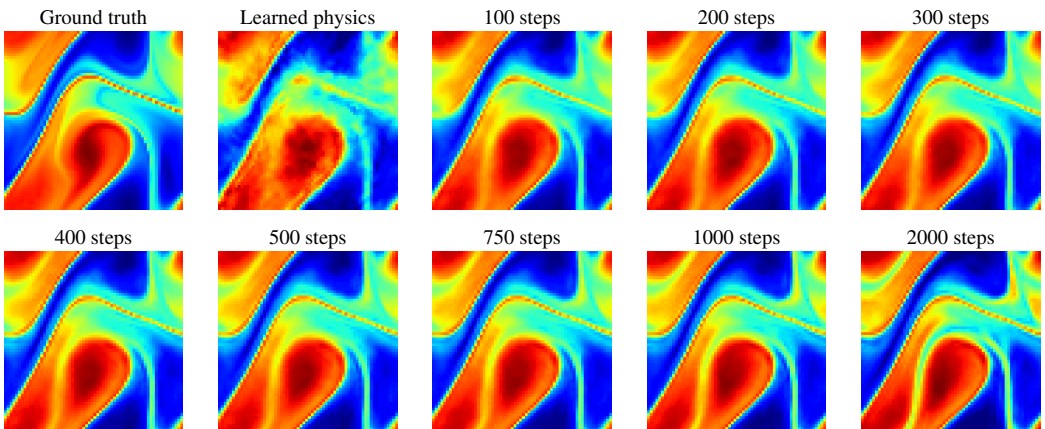

**Figure 24:** Steps with Langevin dynamics for $\epsilon = 2 \times 10^{-5}$ and an additional step with $\Delta t \cdot s_\theta(\mathbf{x}, t)$ which smoothes the images.

## H  ISOTROPIC TURBULENCE

For the learned physics network, we employ a FNO neural network with batch size 20. We train the FNO for 500 epochs using Adam optimizer with learning rate $10^{-3}$, which we decrease every 100 epochs by a factor of 0.5. We train SMDP-ODE with the ResNet architecture for 250 epochs with learning rate $10^{-4}$, decreased every 50 epochs by a factor of 0.5 and batch size 6.

### H.1  REFINEMENT WITH LANGEVIN DYNAMICS

We do a fixed point iteration at a single point in time via:

$$\mathbf{x}_t^{i+1} = \mathbf{x}_t^i + \epsilon \cdot \nabla_{\mathbf{x}} \log p_t(\mathbf{x}_t^i) + \sqrt{2\epsilon} z_t, \tag{60}$$

for a number of steps $T$ and $\epsilon = 2 \times 10^{-5}$ as a post-processing and refinement strategy, cf. figure 23 and figure 24. This is motivated by established methods in score-based generative modelling (Welling & Teh, 2011; Song & Ermon, 2019). For a prior distribution $\pi_t$, $\mathbf{x}_t^0 \sim \pi_t$ and by iterating equation (60), the distribution of $\mathbf{x}_t^T$ equals $p_t$ for $\epsilon \to 0$ and $T \to \infty$. There are some theoretical caveats, i.e. an additional Metropolis-Hastings update needs to be added in equation (60) and there are regularity conditions (Song & Ermon, 2019).

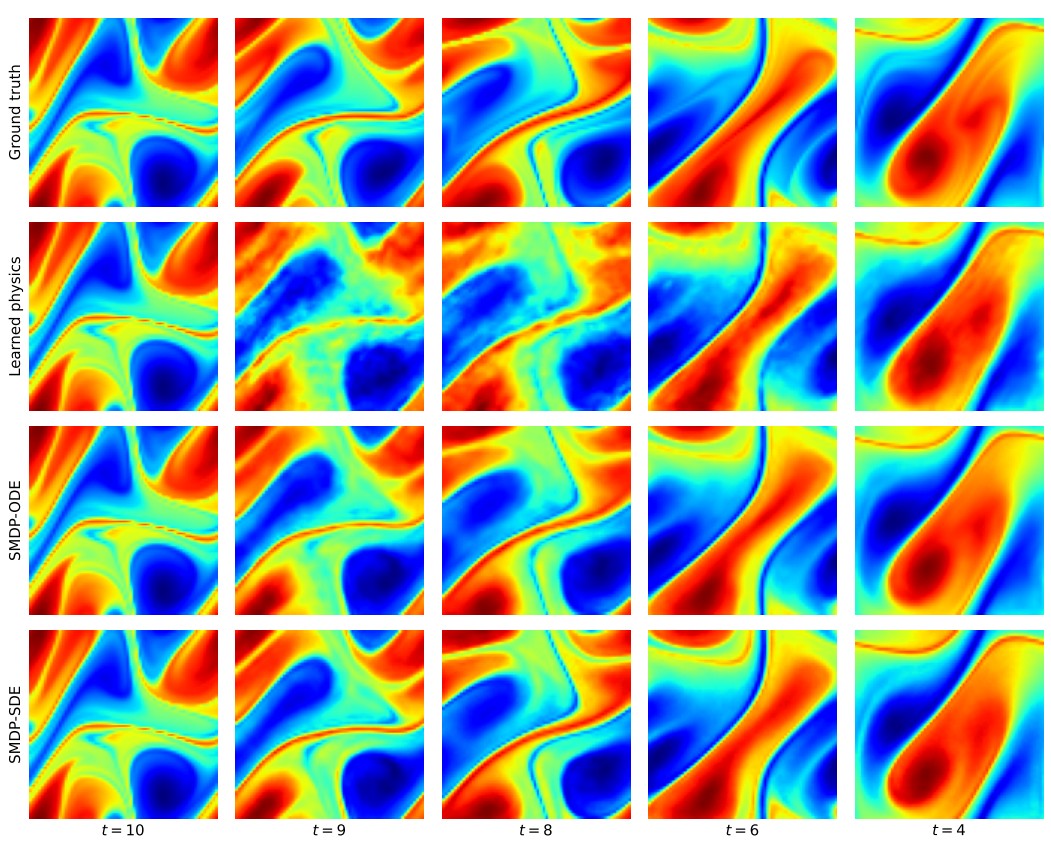

**Figure 25:** Predictions for isotropic turbulence (example 1 of 2).

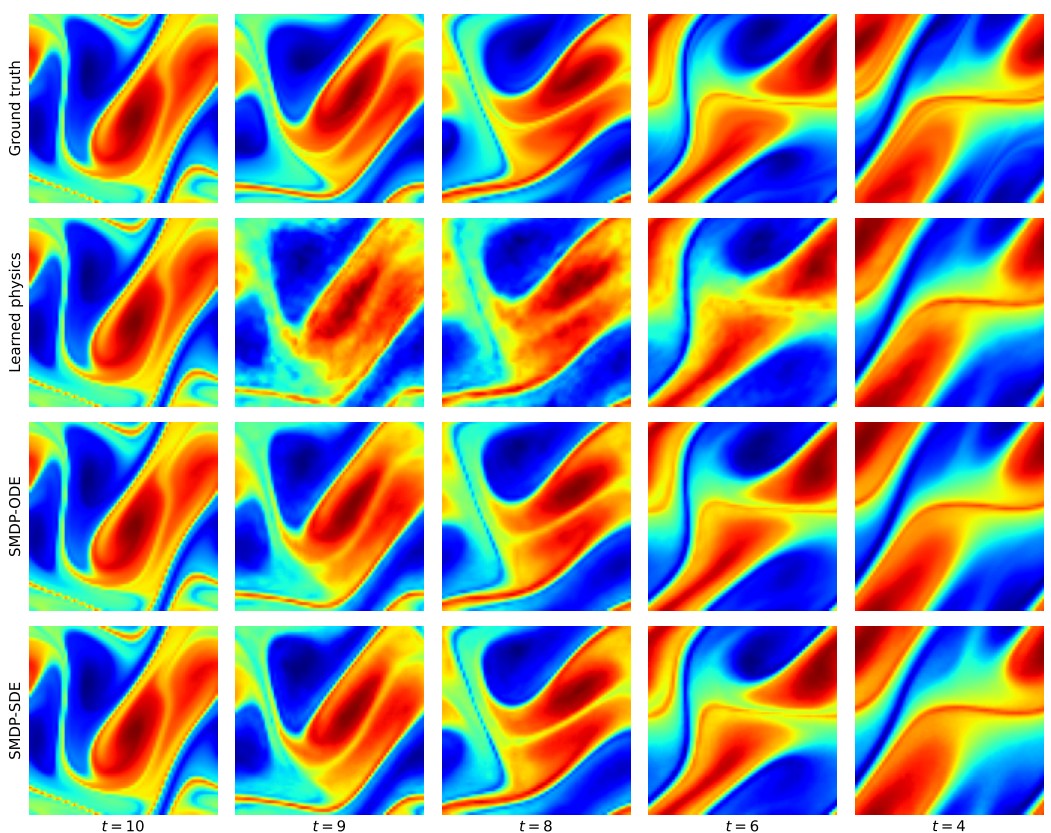

**Figure 26:** Predictions for isotropic turbulence (example 2 of 2).

