# OpenReview forum: "Score Matching via Differentiable Physics"
_ICLR.cc/2023/Conference — Submitted to ICLR 2023_

### Official Review · Reviewer_wxSJ · 2022-10-16

**Confidence:** 4
**Correctness:** 3
**Technical Novelty And Significance:** 2
**Empirical Novelty And Significance:** 3
**Recommendation:** 5

**Clarity, Quality, Novelty And Reproducibility:**

## Clarity

I found the proposed training method to be very unclear. Although the authors spend a long time discussing the objective of denoising score matching, as far as I can tell this is not actually the objective they use. However, their exact objective does not appear to be explicitly stated. They do show a loss $\mathcal{L}((\hat{x}_t), (x_t)) = ((\hat{x}_t) - (x_t))^2$ but $t$ is never defined, the actual definition of $\hat{x}$ and it's initial conditions are never stated, and in general the notation is quite confusing (e.g. are these supposed to be vectors or scalars?). The "bidirectional training" procedure is also never formally described, and neither is the "SMDP-ODE+noise" variant discussed in section 4.1.

*Update: The implementation of their approach and the actual training method used has been clarified substantially. However, I still think that the probabilistic interpretation of their model and the connections to score matching are very unclear.*

## Quality

It is difficult to fully assess the quality of the proposed approach without a clear statement of the training procedure. However, based on my understanding, the objective appears not to be based on score matching at all, and it seems incorrect to refer to their technique as a score matching approach. In particular, the proposed objective seems to be based on L2 distances of sampled trajectories. This is not how score-matching objectives are trained; score matching models are trained to approximate the gradients of the data distribution (as the authors discuss in the background equation (3)). Minimizing the L2 difference between sampled trajectories and learned trajectories that use a score-based parameterization is not the same as learning the score, and I do not think that the learned "score" function will necessarily have anything to do with the gradients of the perturbed data distribution. (The authors state that their loss is "in line with previous work". But although it does resemble the differentiable physics loss used by Um et al., it does not actually seem to be line with the loss in Zhang & Chen, or any other diffusion model I am aware of.)

Similarly, although it is true that the probability flow ODE and the diffusion SDE have the same marginal probability densities, it is not accurate to treat paths computed under the ODE flow as trajectories of the diffusion SDE, or even as an "average" of candidate trajectories. I don't think that minimizing the L2 difference between the probabilistic-flow ODE path and a concrete trajectory sample has any probabilistic interpretation, and so it seems incorrect to refer to this as representing a posterior distribution, or to the SMDP-SDE drawing samples from this posterior distribution.

Most of the experimental results seem high quality from a perspective of evaluating the ability of their model to solve an inverse problem. However, there were additional confusing and seemingly incorrect uses of the diffusion SDE and score matching throughout. For instance, the "Diffusion" baseline doesn't appear to be trained using a diffusion model loss, but instead using their L2 norm loss (is this true?). In section 4.2, the authors introduce a number of heuristic changes which don't make sense under their diffusion process assumption in equation (5) or for a standard score-matching model, such as adding i.i.d. noise to simulation states (instead of Brownian noise) and "decoupling" the guiding function and physics operator. Additionally, they discuss multiple variants using the "score" terminology despite this not having a relationship to the gradient of the data distribution.

*Update: It seems that there is actually a way to interpret this as a score-matching approach (see my discussion with the authors below), but this isn't explained very well, so this is more of a clarity issue than a quality issue.*

## Novelty
Inverting a physical system by using a backward PDE combined with a learned correction term seems novel to me, although it's possible I am unaware of some recent work here. Inferring the initial state is in some ways similar to the setting of [Frerix et al. (2021)](http://proceedings.mlr.press/v139/frerix21a/frerix21a.pdf) but there are also many differences. One very recent diffusion-based approach that incorporates physical system knowledge is [Wu et al. (2022)](https://arxiv.org/abs/2209.00865v1), but the motivation is to add inductive biases to the generative process, not to invert a particular physical system.

The authors claim this is the first work that leverages the reverse-diffusion theorem to solve inverse problems, ~~but as noted above I'm not convinced this method is actually using the reverse-diffusion theorem in a correct way.~~ (I'm not aware of any other works that have done this either, though.)

## Reproducibility
I do not think there are sufficient details about the training algorithm to reproduce this method. (The authors state that they will make the source code available after acceptance but it's not clear how reproducible their experiments will be.)

*Update: I think enough details have been provided in the new revision to make it possible to reproduce this method now.*

**Strength And Weaknesses:**

Strengths:
- Using a learned model to guide a differentiable simulator is a good idea, and using noise to train it seems reasonable as well.
- Their proposed model does well on a variety of experiments, and in particular does better than either directly learning everything with a neural network, or simply running the simulator backwards.

Weaknesses:
- ~~The paper gives very few details about how the model is actually trained. Many details are missing regarding the actual loss used, the distribution of samples in the training set, what is fixed and what is learned, etc. The authors briefly mention "bidirectional training" but it is never described what this actually means.~~
- ~~Based on my understanding of the training procedure, it is not actually trained via score matching, the typical way generative diffusion models are trained. This is why it is necessary to differentiate through the physics; I do not believe such differentiation would be necessary using an actual score-matching objective.~~
  - *After discussion with the authors, it does seem like their objective can be viewed as an approximation of score matching. However, this approximation seems to depend on the "sliding window" used during training being small, and this requirement isn't discussed very clearly in the paper (although there's some discussion in the appendix).*
- ~~The authors appear to be mis-using the relationship between the diffusion SDE and probability flow ODE. In particular, running the probability flow ODE should *not* be interpreted as producing a trajectory sample. The probability flow ODE is a continuous flow such that the marginal distribution at each time is correct, but any two points along a given flow path are not necessarily paths from the same trajectory. However, the authors discuss using the probability flow ODE to generate a "solution" trajectory, and appear to train their model so that this trajectory is close (in L2) distance to entire sampled trajectories. I similarly do not think it is appropriate to treat samples from the SDE as being drawn from a posterior, given that the training process does not seem to be based on KL divergence (or any other mechanism for fitting a sensible posterior over trajectories).~~
  - *After discussion with the authors, it seems like the L2 distance objective can be seen as a maximum-likelihood for a different probabilistic model (not clearly described in the paper right now, but only mentioned in the appendix). And for a small enough sliding window, training this second model would require learning the correct score function, which could then be used to draw samples from the reverse SDE despite their objective not being a KL divergence with the reverse SDE..*
- ~~Overall there are multiple choices that seem very questionable from the perspective of probabilistic generative modeling.~~
- After discussion with the authors, the main remaining weakness I see is that the connections to score matching are not very well explained, and that some details which are presented as minor computational efficiency improvements are actually necessary modeling choices that enable their model to approximate the true score function.
  - The training process actually uses a small window around parts of the trajectory. This is briefly mentioned as an implementation detail in the new "Training and Inference" section, but after reading the appendix and discussing this with the authors, it seems that the small window is actually necessary for their model to recover the true score function. If this window is not used, the learned "score" function might differ substantially from the true score, and the connection to the real SDE/ODE might break down.
  - The authors refer to their objective as a maximum-likelihood bound for the probability flow ODE, but after discussion it appears that the objective is actually a maximum-likelihood bound for a *different* probabilistic model, involving additional Gaussian noise perturbations (which the authors motivate as modeling the numerical inaccuracy of the ODE solver). This perturbed model is not explicitly described in the current submission, except for a partial description in the appendix.

**Summary Of The Paper:**

This paper describes an ODE-based model for inverting the time-evolution of a physical system, aiming to predict a distribution of initial states which could have given rise to this system. Their approach assumes access to a differentiable simulator of the physical system, and represents the reverse-time evolution of the system as either a PDE or an ODE that combines the simulator with a neural network force term. The paper proposes training this force term such that the reversed trajectories under the ODE are close (under L2 norm) to true (forward) trajectories generated by the physical system running in the normal direction. The authors present results for a few physical processes, including heat diffusion, buoyancy-driven flow, and isotropic turbulence, and find that their method yields good approximations of the initial state.

~~Although this approach is motivated by score matching and denoising diffusion generative models, and the authors spend considerable time explaining the background of these models, the actual proposed approach deviates in a number of ways from proper SDE-based diffusion approaches. It seems to me that this is not really a diffusion model in the sense used by the generative model community, and the probability-flow ODE appears to be used "incorrectly" from the perspective of score matching and distribution fitting. I think the proposed methodology would be better understood as a (empirically successful) parameterization and denoising-based training strategy for an inverse model, without reference to generative diffusion models or probability flow.~~

*Updated summary:* After discussion with the authors, it does seem like there is a connection to proper SDE-based diffusion, but this connection is not very well explained. In particular, while their training objective isn't exactly a maximum-likelihood objective for the SDE itself or for the probability-flow ODE, it can be seen as a maximum-likelihood objective for a different, perturbed ODE flow model. And if the "sliding window" used during training is sufficiently small, the optimum of their objective will approximate the score function of the original SDE/ODE. I still feel that these connections aren't explained very well in the current revision of the paper.

**Summary Of The Review:**

This paper has some interesting ideas for using learned ODEs to find the initial states of a physical system conditioned on a final state, and the method seems to work well empirically. However, I think in its current form the paper makes many unjustified and incorrect claims about the relationship between this model and SDE-based score-matching diffusion generative models. The paper also omits many important details about the training procedure, which seems substantially different than prior work with diffusion models.

~~As such I do not think the paper should be accepted in its current state. I would be willing to raise my score if the authors clearly explain what their actual training algorithm is and remove the misleading claims about the relationship to score matching and probabilistic inference (or, alternatively, if they can justify the sense in which their objective is equivalent to score matching and explain why my concerns are unfounded).~~

*Updated review:* The authors have added the missing details about the training procedure, which is now much better explained. They have also added a derivation which claims their objective is a probabilistic lower bound that is (indirectly) connected to score matching. I have some issues with the way this is presented in the paper, but after discussion with the authors (see below) I've been convinced that there is indeed a sense in which the actual objective used (for a sufficiently small sliding window) could be viewed as approximately doing score matching. However, this connection and the dependence on the window size are not very well explained in the paper, and there are still some misleading claims about maximum-likelihood training. I have raised my score from 3 to 5, since I no longer think there are critical problems, but still I'm hesitant to recommend acceptance due to these remaining clarity issues.

---

> ### Author Response · Authors · 2022-11-18
> **Response to Reviewer wxSJ**
>
> We thank the reviewer for the assessment and for the detailed questions regarding the proposed methodology.
>
> **Missing details about model training, loss, distribution of samples and bidirectional training**
> We agree with the reviewer that the presentation of the training details and loss was unnecessarily short. As stated in the paper, our implementation is similar to Um et al. (originally for learned correctors in physical simulation where dynamics are unrolled for multiple steps) and Zhang & Chen (minimizing the difference between trajectories of the forward SDE and trajectories of the reverse-time SDE plus an additional term for the likelihood of the trajectory end point). We have updated the paper and expanded the appendix with additional information about the implementation of our method and differences with previous methods.
>
> **Training procedure is not based on score matching the typical way generative diffusion models are trained** Our methodology is fundamentally different from denoising score matching, where dynamics are described by a corruption process, whereas we consider arbitrary physics operators. Therefore, we do not have access to the transition kernel at training time. In the appendix of the updated paper, we have included additional theoretical justifications and show that our loss based on the L2 distance optimizes a variational lower bound for maximum likelihood training of the CNF corresponding to the probability flow ODE. We refer to previous work as a reference to how maximum likelihood training is a valid method to optimize the score matching objective (Song et al. (2021), Huang et al. (2021)). Implementation-wise, our training and loss are similar to Zhang & Chen, but there are non-trivial differences and we provide an extended discussion in appendix A. Zhang & Chen consider generative modeling problems, where one of the distributions is approximately Gaussian. They minimize the difference between trajectories from the forward SDE and reverse-time SDE to learn the score and drift. An important difference is that our method fits the deterministic probability flow ODE trajectories to the non-deterministic SDE paths.
>
> **Differentiation through the physics would not be necessary using an actual score-matching objective?** Our methodology can also be applied when we unroll the dynamics for \$N=1\$ steps. We include additional experiments in the appendix regarding the reconstruction MSE of the heat equation, where we change the number of steps for which we unroll the dynamics and we empirically observe that longer rollouts of the dynamics yield improved model performance. For the reverse-time SDE, interactions between ill-conditioned physics and noise are very difficult to control when only considering single step updates during training. As an additional empirical verification of our method, we have included a toy experiment in appendix B, where we directly compare our method to Implicit Score Matching. The corresponding comparison of the learned score functions shows that our method is in good agreement with the score learned by Implicit Score Matching. This example also shows that our method correctly captures the posterior distribution. In fact, it outperforms ISM in terms of “physical accuracy”.
>
> **Relationship between probability flow ODE and SDE. Training is not based on KL divergence?**
> We agree with the reviewer that a trajectory from the probability flow ODE is not the same as a trajectory from simulating the SDE, as both have different path measures. To avoid confusion, we have revised all parts of the paper that could be mistaken to claim otherwise. Nonetheless, the probability flow ODE trajectory has certain desirable properties, such as transforming a sample from the distribution \$p_0\$ to \$p_T\$ and we can include trajectories from the probability flow ODE in the experiment evaluation. As mentioned above, we provide additional theoretical justification based on the maximum likelihood training of CNFs in appendix A, why our objective is valid and treating samples from the reverse-time SDE with our learned score as samples from the posterior is appropriate.
>
> **Comments about Clarity and Quality** We have clarified the definition of the loss function, bidirectional training and the SMDP-ODE+noise variant. The updated paper contains additional information about the training implementation in the appendix. We believe that we have addressed the open questions in the methodology in our previous points. We have removed statements about the probability flow trajectory being an “average” of candidate trajectories.
>
> Please let us know if there are any remaining issues, we’d be happy to provide further details.
>
> References:
> -   Song et al. (2021): Maximum Likelihood Training of Score-Based Diffusion Models
> -   Huang et al. (2021): A Variational Perspective on Diffusion-Based Generative Models and Score Matching
> -   Zhang & Chen (2021): Diffusion Normalizing Flow

---

> > ### Author Response · Authors · 2022-11-18
> > **Response to Reviewer wxSJ (continued)**
> >
> > We post answers to additional comments by the reviewer here:
> >
> > **Training of “Diffusion” baseline using L2 norm in the heat equation experiment** The “Diffusion” baseline is based on Rissanen et al. (2022). When only unrolling one step of the dynamics, their loss is equivalent to our L2 loss, see Equation (10) in their paper and the discussion directly below. In the same paragraph the authors explain the connection of their loss to denoising score matching. Nonetheless, we agree with the reviewer, that the label “Diffusion” can be confusing here, so we updated it in the revised paper.
> >
> > **Adding i.i.d. noise to simulation states (instead of Brownian noise)** We also agree that adding i.i.d. noise in section 4.2 for each simulation state is a deviation from the SDE perspective, i.i.d. noise is highly relevant for many practical applications. Evaluating whether the probability flow ODE and reverse-time SDE still produce plausible solutions in such a case is an important question that we try to answer. A different perspective on this is that while for the time from \$0\$ to \$T\$, the noise is not Brownian, it can still be treated locally as Brownian from one timestep to the next. As our method does not make any specific assumptions about the simulation start and end time, we can think of using i.i.d noise in terms of using multiple smaller score models and SDEs for time \$t_i\$ to time \$t_{i+1}\$ that are chained together. The conditions for the reverse diffusion theorem are then satisfied for smaller time intervals (but not the entire simulation from time \$0\$ to \$T\$).
> >
> > **Decoupling of the “guiding function” and physics operator** The decoupling of the score and physics as described in the paper, does not have an impact on the theoretical side. However, we believe that when time steps are too large and the physics dynamics are strong, it is easier to train and obtain stable trajectories when decoupling the score and physics. For our method to learn the exact score, we would require that the time steps become infinitely small, so we believe this tradeoff is reasonable. We have included both of these viewpoints in the paper.
> >
> > References:
> > -   Rissanen et al. (2022): Generative Modelling with Inverse Heat Dissipation

---

> > > ### Comment · Reviewer_wxSJ · 2022-11-20
> > > **Remaining concerns**
> > >
> > > Thank you for adding the additional details about the training objective and loss function; it is much easier now to understand what the proposed method is.
> > >
> > > I have looked through the new section in the appendix, which discusses the connection between the proposed objective and score matching. At a high level, the argument appears to be:
> > >
> > > 1. Our ultimate goal is to invert the forward SDE and learn the score function.
> > > 2. To do this, it is sufficient to match the marginals of the inverse SDE and forward SDE, because both of these are connected through the CNF/ODE perspective.
> > > 3. We can do this in principle by training the model as a sort of family of CNFs: for each time $t$, we make sure that the learned probability flow's marginals at time $t$ match the data distribution marginals at time $t$, when initialized at an independent random sample of the value at time $t+1$. (However, this is difficult to do exactly, because we don't have densities for time 0 or for time T, so we can't just apply the change-of-variables formula as for a normal ODE flow model.)
> > > 4. To circumvent these difficulties, a different loss is used, which minimizes L2 differences between samples from the true forward SDE and solutions from the reverse ODE. This is stated to be a variational bound on the desired marginal-matching objective in 3.
> > >
> > > Unfortunately I'm still not convinced that the derivation in the appendix actually justifies the algorithm used. Some of the things that do not seem right to me:
> > >
> > > - The derivation seems to depend on an approximation of the (deterministic) ODE mapping between two timesteps $t$ and $t+1$ as a conditional Gaussian with some numerical error, and tries to maximize a likelihood that a noisy computation of the ODE yields a particular solution. This seems quite strange to me. If the performance of the method *requires* the ODE to be computed less accurately in order to be mathematically sound, that seems like an issue with the method, and it's not really talking about the ODE anymore, just some particular implementation of it. Furthermore it's unlikely that the numerical error is exactly Gaussian for any particular ODE solver, calling the variational bound interpretation into question.
> > > - In Equation 39, the authors introduce a "normalizing constant" which is based on truncating a distribution to a particular high-probability set. Then in equation 44 this constant is dropped. However, I think that the truncation depends on the model parameters (because it depends on $\tilde{I}$ which depends on $\tilde{S}$ which depends on $S$ which depends on $p^{ODE}$), so the constant also depends on those parameters. As such it's not justified to remove this term when optimizing the ODE.
> > > - This normalizing constant appears to be used in order to define an alternative proposal distribution $\tilde{p}_t$. But the proposed algorithm doesn't actually seem to use this proposal distribution; in practice they just reuse a (non-independent!) sample from the true marginals $p_t$, or possibly a (also non-independent!) output from their ODE at a different timestep. So I don't know why the properties of $\tilde{p}$ would justify the way their model is actually trained.
> > >   - One particularly odd aspect of this: although the method itself is based on differentiating through the learned ODE, it seems that the theoretical justification in the appendix seems to require NOT doing that but instead using a different distribution of samples based on this truncated marginal distribution. So this doesn't really justify differentiating through the ODE, as far as I can tell, which is one of the main ways this method differs from prior work.
> > > - Between equations (45) and (46) it appears that the likelihood bound differs from their actual objective by a value proportional to (1 / machine epsilon), which could be extraordinarily large.
> > >
> > > Regarding DiffFlow (Zhang & Chen), although Zhang & Chen's Equation (15) is somewhat similar to the proposed objective, the method proposed here seems much less probabilistically sound. In particular Zhang & Chen use pairs of adjacent samples $(x_t, x_{t+1})$ from the forward process, and evaluate the likelihood of $x_t$ under their model conditioned on $x_{t+1}$, which turns into an L2 distance between the predicted mean and $x_{t}$ and is also directly related to the score function. On the other hand, the method in this work seems to use non-local pairs $(x_t, x_T)$, and evaluate the L2 distance between $x_t$ and a deterministic function of $x_T$, which is only interpretable probabilistically under the odd assumption about Gaussian computation noise and isn't obviously related to the score at all.
> > >
> > > I do appreciate the effort on the part of the authors here, and many of my other comments have been addressed, but I still stand by my original belief that the actual algorithm proposed does not appear to do score matching, and that the claims to that effect are not properly justified.

---

> > > > ### Author Response · Authors · 2022-11-24
> > > > **Response to remaining concerns**
> > > >
> > > > We'd like to thank the reviewer for the fast reply, and additional, detailed comments regarding the new derivation of our method. We believe there were still a few misunderstandings and we give clarifications and corrections where needed, but most importantly, we'd like to emphasize that our method really represents a valid score matching objective, as outlined in more detail below.
> > > >
> > > > **Modeling via deterministic ODEs plus perturbations** The approximation of distributions $p$ as Gaussians with mean centered on the mode of $p$ is very common for variational inference, see for example Bishop (2006), *Chapter 4.4 The Laplace Approximation*. Modeling the uncertainty of numerical solutions as probability distributions is also widely used, e.g. Conrad et al. (2016), *Statistical analysis of differential equations: introducing probability measures on numerical solutions*, Schober et al. (2019), *A probabilistic model for the numerical solution of initial value problems*, or Kersting et al. (2020) *Differentiable Likelihoods for Fast Inversion of 'Likelihood-Free' Dynamical Systems* for Gaussian ODE filtering. Note that the derived loss function for our method does not depend directly on the variance of the Gaussian. It would also be inaccurate to say that our method requires numerical solutions to have errors, as the approximation with Gaussians allows for the use of variational methods to optimize the objective, even if the solutions are exact.
> > > >
> > > > **Equivalence to score matching objectives for single steps** First, we'd like to clarify the relation of our method to score matching for single steps, i.e. from time $t_{i}$ to $t_{i+1}$ for which an approximation via Euler Maruyama steps is common practice. Here, the relation between minimizing the L2 distance between the ODE trajectories and the score matching objective becomes obvious when considering the KL divergence between the joint distribution $p_{t_i} \times p_{t_{i+1}}$ and the distribution induced by the ODE. Recall from the paper, that the distribution $p_{t_{i}}^{\mathrm{ODE}, t_{i+1}}$ samples from $p_{t_{i+1}}$ and computes the probability flow ODE for $t_i$ with the network $s_\theta$. In the equation below, with a slight abuse of notation, we treat it as a distribution for the joint probability. Then, when considering the objective of the bidirectional training (in both time directions) for a single step, we obtain
> > > > $$ \mathrm{KL}(p_{t_i} \times p_{t_{i+1}}||\, p_{t_{i}}^{\mathrm{ODE}, t_{i+1}}) + \mathrm{KL}(p_{t_{i+1}} \times p_{t_{i+2}}||\,p_{t_{i+2}}^{\mathrm{ODE}, t_{i+1}}) $$
> > > > $$ = E_{(x_{t_i}, x_{t_{i+1}}, x_{t_{i+2}}) \\sim p_{t_i} \\times p_{t_{i+1}} \\times p_{t_{i+2}}}
> > > > \left[\log \frac{p_{t_i}(x_{t_i}|x_{t_{i+1}})p(x_{t_{i+1}})}{p_{t_i}^{\mathrm{ODE},t_{i+1}}(x_{t_i}|x_{t_{i+1}})p(x_{t_{i+1}})} + \log \frac{p_{t_{i+1}}(x_{t_{i+2}}|x_{t_{i+1}})p(x_{t_{i+1}})}{p_{t_{i+2}}^{\mathrm{ODE},t_{i+1}}(x_{t_{i+2}}|x_{t_{i+1}})p(x_{t_{i+1}})} \right] $$
> > > > $$ = E_{x_{t_{i+1}}}\left[\mathrm{KL}(p_{t_i} | x_{t_{i+1}} ||\,p_{t_{i}}^{\mathrm{ODE}, t_{i+1}} | x_{t_{i+1}} ) + \mathrm{KL}(p_{t_{i+2}} | x_{t_{i+1}} ||\,p_{t_{i+2}}^{\mathrm{ODE}, t_{i+1}} | x_{t_{i+1}} ) \right] $$
> > > >
> > > > For small time steps, we can use the approximations using the Euler Maruyama steps for $p_{t_i} | x_{t_{i+1}}$ (reverse-time SDE), and $p_{t_{i+2}} | x_{t_{i+1}}$ (forward SDE) and additionally model the corresponding ODE solution as Gaussians with small standard deviation $\epsilon > 0$. Then, for example the mean of $p_{t_{i+2}} | x_{t_{i+1}}$ is $x_{t_{i+1}} + (t_{i+1}-t_i)\mathcal{P}(x_{t_{i+1}})$ and the mean of $p_{t_{i+2}}^{\mathrm{ODE}, t_{i+1}}|x_{t_{i+1}}$ is $x_{t_{i+1}} + (t_{i+1}-t_i)[\mathcal{P}(x_{t_{i+1}}) - 0.5 g_{t_{i+1}}^2s_\theta(x_{t_{i+1}}, t_{i+1})]$. Therefore, the KL divergence can be computed directly for multivariate Gaussians and is given by
> > > > $$
> > > > 	    \Delta t \frac{1}{\epsilon^2} g_{t_{i+1}}^4 E_{x_{t_{i+1}}}[|| \nabla_{x_{t_{i+1}}} p_{t_{i+1}}(x_{t_{i+1}}) - \frac{1}{2}s_\theta(x_{t_{i+1}}, t_{i+1}) ||^2_2 + ||\frac{1}{2}s_\theta(x_{t_{i+1}}, t_{i+1})||^2_2] + C,
> > > > $$
> > > > with $C$ independent of $\theta$. For any $\epsilon > 0$, this is minimized by
> > > > $$
> > > > 	    \frac{1}{2}s_\theta(x_{t_{i+1}}, t_{i+1}) = \frac{1}{2}\nabla_{x_{t_{i+1}}}\log p_{t_{i+1}}(x_{t_{i+1}}).
> > > > $$
> > > > Thus, $s_\theta(x_{t_{i+1}}, t_{i+1}) = \nabla_{x_{t_{i+1}}} \log p_{t_{i+1}}(x_{t_{i+1}})$, which **directly** links our method and derived loss for a single step of unrolling the dynamics to the score matching objective.
> > > >
> > > > You are right that this requires to model the conditional ODE distribution as a Gaussian. We believe this is a valid starting point for linking physics simulations and score matching (to be improved in the future), and it provides a simple loss that can be optimized without the need for a specific choice of  $\epsilon$.

---

> > > > > ### Author Response · Authors · 2022-11-24
> > > > > **Response to remaining concerns (continued)**
> > > > >
> > > > > **Role of the differentiable solver**
> > > > > While our derivation focuses on the score matching objective for single steps, and incorporates the extension to sequences with the CNF viewpoint,
> > > > > we have observed empirically that the performance improves substantially when unrolling multiple steps.
> > > > > For example, even with a lot of training data for a toy problem (Appendix B), Implicit Score Matching (ISM) has difficulty producing good reverse-time SDE trajectories. ISM does not integrate the interaction between the neural network and the physics operator.
> > > > > We found that for successful learning the temporal dynamics should be included in the training as well.
> > > > > Extending the theory to these tough, generic cases is beyond the scope of this work, and an interesting topic to follow up on.
> > > > >
> > > > > **Normalizing constant in equation (44)**
> > > > > The dropping of the normalizing constant in equation (44) is justified because with assumption (iii) it does not depend on $\theta$. The set $S_\kappa$ depends on $p^{\mathrm{ODE}}$, but only in the sense that we model it as a Gaussian with some mean $x$ and variance.
> > > > >
> > > > > **Remaining concerns regarding the sampling**
> > > > > Thank you for pointing out the problems in our argumentation regarding sampling.
> > > > > We believe that we can address these concerns quite easily, which in turn  simplifies the derivation. In particular, continuing at maximizing equation (20), we can write
> > > > > $$
> > > > > 	    E_{x_T} \left[ \log p_T^{\mathrm{ODE},0}(x_T) \right]
> > > > > $$
> > > > >
> > > > > $$
> > > > > = E_{x_T} \left[ \log \left( E_{x_0} \left[ p_T^{\mathrm{ODE},0}(x_T| x_0) \right] \right) \right]
> > > > > $$
> > > > >
> > > > > $$
> > > > > = E_{x_T} \left[ \log \left( E_{x_0|x_T} \left[ \frac{p_0(x_0)}{p_0(x_0|x_T)} p^{\mathrm{ODE},0}_T (x_T| x_0) \right] \right) \right]
> > > > > $$
> > > > >
> > > > > $$
> > > > > \geq E_{x_T} E_{x_0|x_T} \left[
> > > > > 	    \log \left( \frac{p_0(x_0)}{p_0(x_0|x_T)} p_T^{\mathrm{ODE},0}(x_T| x_0) \right)  \right]
> > > > > $$
> > > > >
> > > > > $$
> > > > > = E_{x_T} E_{x_0|x_T}  \left[ \log \left( \frac{p_0(x_0)}{p_0(x_0|x_T)} \right) + \log \left( p^{\mathrm{ODE},0}_T(x_T| x_0) \right) \right],
> > > > > $$
> > > > >
> > > > > which is the same as maximizing
> > > > >
> > > > > $$
> > > > > 	    E_{x_T} E_{x_0|x_T} \left[ \log \left( p_T^{\mathrm{ODE},0}(x_T| x_0) \right) \right].
> > > > > $$
> > > > >
> > > > > The derivation of this variational lower bound is not based on modeling $p_T^{\mathrm{ODE},0}(x_T| x_0)$ as a Gaussian and works for any two time steps $t_i$ and $t_j$ instead of $0$ and $T$.
> > > > >
> > > > > **Extension to multiple steps and training stability** The maximum likelihood objective of our method is to maximize $E_{x_0} \left[ \log p_0^{\mathrm{ODE},T}(x_0) \right]$. This requires solving the ODE for many steps, making the optimization more difficult and potentially unstable. Therefore, we optimize the objective not only for $\mathrm{KL}(p_0||p_0^{\mathrm{ODE}, T})$, but for all  $\mathrm{KL}(p_{t_i}||p_{t_i}^{\mathrm{ODE}, t_j})$. Using the Laplace approximation for the ODE solution, this can be optimized via the L2 distance, i.e. for our implementation we sample a trajectory $(x_0, ..., x_T)$ and compute the sum of these losses as the L2 distance between the ODE trajectory and the SDE trajectory similarly to the learned corrector approaches for physical simulations.
> > > > >
> > > > > Note that as argued above, for single time steps, this can be viewed as directly optimizing the score matching objective, while for longer unrolling of dynamics, our method optimizes a variational lower bound on the likelihood and can be seen as maximum likelihood training with established connection to score matching. As the sliding window method in our implementation optimizes a combination of both, we believe that making a clear connection between the proposed methodology and score matching is justified.

---

> > > > > > ### Comment · Reviewer_wxSJ · 2022-11-24
> > > > > > **Discussion (continued)**
> > > > > >
> > > > > > Apologies for my disjointed reply, I did not see this continued comment until after posting my other reply above.
> > > > > >
> > > > > > Is the sliding window supposed to be part of the proposed model itself, rather than the training algorithm? If so, that was very unclear from the paper. If I'm understanding what you're saying here, it seems that the probabilistic model you are actually working with is something like
> > > > > >
> > > > > > $p\_{t\_1 | t\_2}^{ODE}(x\_{t\_1} | x\_{t\_2}) = \\mathcal{N}( f\_{t\_1 | t\_2}^{ODE}(x\_{t\_2}), \\epsilon^2 (t\_2 - t\_1)^2 )$
> > > > > >
> > > > > > where $f\_{t\_1 | t\_2}^{ODE}$ is the function that solves the ODE starting at time $t_2$ and running backward to $t_1$ (e.g. over the sliding window), parameterized by $\theta$, and $\\mathcal{N}$ denotes a Gaussian distribution. Then you take exact samples $(x\_{t\_1}, x\_{t\_2}) \sim p$ from the real distribution, and maximize the likelihood of $x\_{t\_1}$ under your model conditioned on $x\_{t\_2}$?
> > > > > >
> > > > > > Then perhaps what you are saying is that, if we take the limit $t\_2 - t\_1 \to 0$ (the "single step" case), we end up with a score-matching objective on $s_\theta$. On the other hand, if we set $t\_2 = T$, we are no longer really doing score matching, but we can still interpret it as max-likelihood training of the sliding-window version of the model (which incorporates the explicit noise term $\epsilon$).
> > > > > >
> > > > > > If I'm understanding this correctly, this perspective does seem more principled, but I don't think any of this is explained in the paper. Importantly, the model I've written above is not the same as the reverse SDE in (2) or the CNF in (3), which is the main focus of the paper right now.

---

> > > > > > > ### Author Response · Authors · 2022-11-25
> > > > > > > **Discussion**
> > > > > > >
> > > > > > > Thank you for the fast reply and the detailed comments. Yes, we think your general understanding is correct, but we want to clarify two aspects below.
> > > > > > >
> > > > > > > **Connection to reverse-time SDE equation (2)**
> > > > > > > The model in your comment does not correspond to the reverse-time SDE, as you have pointed out. However, we still make use of the connection between the reverse-time SDE and the CNF to generate samples with SMDP-SDE, for which we just plug the learned network $s_\theta$ into the reverse-time SDE. In theory, if $s_\theta$ is identical to the score, this allows for sampling from the posterior and reverses the data generation process. Also in that case, it is a minimizer for the KL divergence between the actual distribution $p_0$ and the distribution induced by $p_0^{\mathrm{ODE},T}$ (the KL divergence would be $0$). Thus, it would represent an optimal solution for the maximum likelihood training of the CNF.
> > > > > > > The important point here is that the connection between the data generation, posterior sampling and CNF is still of central importance for our paper.
> > > > > > >
> > > > > > > **Connection to CNF equation (3)**
> > > > > > > Regarding the connection to the CNF equation (3), for a practical optimization during training, we slightly perturb the otherwise completely deterministic trajectories. If these perturbations are small, the model we obtain from these perturbations is still very close to the deterministic CNF, but can be optimized much more easily. The perturbed model would then correspond to the model from your comment.
> > > > > > >
> > > > > > > We hope that we've addressed the concerns of your original review with these answers. We will definitely update the main body of our work to reflect these unclear points and our discussion. E.g., we will point out in section 3 that we rely on perturbations in equation (3) and use the sliding window approach for more stable training, which is currently discussed as part of the more detailed training setup in the revised version of the paper. However, we also want to stress again that these changes leave the core of our method and all results/experiments unmodified, as the clarifications are only about how we establish the relationship between learning via physics simulations and the score matching/maximum likelihood objective.

---

> > > > > > > > ### Comment · Reviewer_wxSJ · 2022-11-25
> > > > > > > > **Response and updated review**
> > > > > > > >
> > > > > > > > Hm, right, if you're able to learn the score function then I suppose you can reconstruct the reverse-time SDE and CNF even if your objective is not directly a KL divergence on either the SDE or the CNF.
> > > > > > > >
> > > > > > > > I've updated my review and increased my score from 3 to 5. Based on this discussion, I agree that there is a sense in which the actual objective used (for a sufficiently small sliding window) could be viewed as approximately doing score matching, and that this doesn't affect the experimental results. I still think the connection isn't explained very well in the paper, and the period for uploading new versions of the paper is closed, so I'm hesitant to recommend acceptance, but I no longer think that there are critical errors.
> > > > > > > >
> > > > > > > > My specific suggestions for the next version of the paper would be:
> > > > > > > >
> > > > > > > > - Explicitly include the sliding window in equation (7) and (9).
> > > > > > > > - Discuss the "single step" case (e.g. when the sliding window is taken as $(t_{i}, t_{i+1})$) explicitly in Section 3, and make it clear that the objective only recovers $s_\theta$ when the sliding window is sufficiently small. (In other words, having a small sliding window doesn't just increase computational efficiency, it also makes the objective closer to score matching.)
> > > > > > > > - Fix the claim "We optimize $s_\theta(x, t)$ via maximum likelihood training of the probability flow
> > > > > > > > ODE, as discussed in section 2. For this, we maximize a variational lower bound for the maximum
> > > > > > > > likelihood objective, which we estimate by minimizing the following loss" in section 3. Based on our discussion, I don't think this is true; it's not maximum-likelihood training of the ODE described in section 2 exactly, but instead maximum-likelihood training of the Gaussian-perturbed sliding-window version of the ODE discussed in my comment above (and also in the appendix).

---

> > > > > > > > > ### Author Response · Authors · 2022-11-28
> > > > > > > > > **Updated review**
> > > > > > > > >
> > > > > > > > > Dear reviewer, thank you very much for acknowledging the connection between score functions and reconstructing the target SDE/ODE of our approach. We’d like to stress that we’re very grateful for the detailed comments, and we will start working on a thorough update including these clarifications right away. We also want to stress that we’d be happy to closely work together with the reviewers when preparing revision to ensure that all these aspects can really be understood and followed by readers outside of our core team.
> > > > > > > > >
> > > > > > > > > We know that it is asking for a lot - nonetheless, we’d like to ask you to consider updating the score to actually recommend acceptance of our work. Right now, a score of 5, unfortunately, still implies a reject. As mentioned above, we can ensure that we will work hard to iron out all unclear parts in the exposition of our submission.

---

> > > > > > > > > > ### Comment · Reviewer_wxSJ · 2022-12-01
> > > > > > > > > > **Response**
> > > > > > > > > >
> > > > > > > > > > My understanding of the ICLR review process is that substantial changes to the paper are not allowed at this stage of the review process, and so my review is based on the revision of the paper as is currently submitted. But I will confirm with the AC to make sure I've understood the process correctly.

---

> > > > > ### Comment · Reviewer_wxSJ · 2022-11-24
> > > > > **DIscussion**
> > > > >
> > > > > **Modeling via deterministic ODEs plus perturbations:** Sure, I understand that in some cases you can model uncertainty of numerical solutions using a Gaussian distribution. But when this is done, my understanding is that this Gaussian distribution is considered to be part of a probabilistic model specified in advance. In contrast, here you specify a probabilistic model (the CNF of the probability flow ODE in equation 3) and claim to be doing maximum likelihood training of it, but the "likelihood" is NOT the likelihood of the data under the model you specify, but instead the "likelihood" under a different model that is only defined inside the proof, and this second likelihood depends on the numerical approximation error as a modeling choice.
> > > > >
> > > > > Perhaps you could formalize your approach as doing maximum likelihood / KL divergence training of a different probabilistic model, which would be something like an underlying deterministic ODE with observation noise at each timestep (similar to a hidden Markov model). But that's not the CNF model defined in (3), and it's also not the SDE model in (2).
> > > > >
> > > > > **Equivalence to score matching objectives for single steps:** I think this is answering the wrong question. Your training objective does not actually train using single steps, right? Based on Section 3, my understanding is that your actual approach integrates out the ODE through time starting at time T (or time 0 for reverse training) and uses that to evaluate your loss. Notably, if we let $x\_{t\_{i+1}}, x\_{t\_{i}}, x\_T \sim p$ be draws from the true distribution, your actual loss does NOT involve evaluating the "score function" model at $x_{t_{i+1}}$ and comparing it to $x\_{t\_{i}}$ (which is what your derivation above does), but instead evaluates it at $x^{ODE}\_{t\_{i+1}} = f^{ODE}\_{t\_{i+1}}(x\_T)$, e.g. the deterministic solution to the ODE that you differentiate through.
> > > > >
> > > > > Overall it's somewhat plausible to me that, if you wrote your probabilistic model in a different way, and evaluated your objective on a different set of points, that it would then correspond to score matching and/or a probabilistically-valid likelihood objective. But I think the rewritten model would not be the one in equation (3), and the new objective would not be the one you discuss in Section 3, so it doesn't seem like this justifies the actual method you propose.

---

### Official Review · Reviewer_N4rm · 2022-10-24

**Confidence:** 3
**Correctness:** 3
**Technical Novelty And Significance:** 3
**Empirical Novelty And Significance:** 4
**Recommendation:** 5

**Clarity, Quality, Novelty And Reproducibility:**

Clarity (Fair): the method section can use a bit of improvement, such as what is the loss function, and some notational changes from background to method section. Also, it would be nice to reiterate how the setting is different from standard diffusion models in generative settings (where we only care about the marginal at t=0).

Quality (High): the experiments are generally well-done, except for the fact that posterior sampling is not well evaluated.

Novelty (High): while this seems to be an application of diffusion normalizing flows, it is an interesting one beyond the scope of the original paper.

Reproducibility (Fair): while the authors promised code release, the method section does not clearly describe the precise loss function. Specifically, it would be nice to expand the line on "We update the weights θ of sθ by unrolling and backpropagating though all ODE solver steps"

**Details Of Ethics Concerns:**

N/A.

**Strength And Weaknesses:**

Strength:
- Empirical performance seems to be better compared to baselines.
- The idea is an interesting take at inverse problem solving using the idea of diffusion models.

Weakness:

The method section is not very clear.
- For example, in $\{x_t^n\} \subset R^{d \times T}$ is confusing. What is $d$? With notation $d \times T$ does this imply that the trajectories are sampled at discrete times? The $t_{m}$ notation should come after you discuss Euler-Maruyama discretization steps?
- What is the loss function, and are there any principled justification to this? My understanding is that this is the loss function in (Zhang & Chen, 2021) but with the drift being non-trainable. If this is true, then it would help to make this explicit, as it helps understanding the paper by a great deal. While the technical novelty is lowered, this is still a non-trivial application of diffusion normalizing flows (since Zhang & Chen mostly considered generative modeling problems).

It is unclear why both ODE and SDE can be used.
- The goal in inverse problems is to sample from the full posterior. In that sense, ODE will only give one solution, which as the paper suggests, does not cover many candidate solutions. This is also because ODE does not recover the measure of the entire path, just the marginals. Wouldn't this suggest that, assuming the forward model is correct, the SDE is the **only** correct way of sampling from the posterior, since it should match the path measures.

Evaluations are not precisely targeted towards posterior sampling.
- If my understanding is correct, then in Table 1, only one path is compared against. However, if we have a non-zero g, then there are naturally many paths with similarly high probability of occurring, and comparing against one path is going to place posterior sampling at a disadvantage. I wonder if it is possible to obtain a "ground truth" posterior and compare path distributions with that? For example, using metrics like maximum mean discrepancy or divergences based on KDEs.
- The paper address the above by having g=0 during evaluation. However, would this generate the same score function as in g=0.1 during training?


Not weakness, but curious questions:
- Why does the FNO perform so badly in Table 1?

**Summary Of The Paper:**

The paper uses score matching SDEs for learning inverse problems with physical simulators. Different from common diffusion models, the forward SDE is changed: the drift is given by a physical model and the diffusion term does not grow to infinity as time increases. Training and inference algorithms based on SDEs and probabilistic ODEs are discussed. Empirical results are evaluated on several inverse problems and compared against baselines, demonstrating the empirical advantage of the proposed SMDP method.

**Summary Of The Review:**

While this seems to be an interesting application of diffusion normalizing flows to physical inverse problems, I am a bit concerned by the clarity in the method section. There are some notation issues, and even the loss function is not precisely defined (either in the main paper or in the appendix). But I do think the method itself is valid and experiments generally support the validity of the method.

---

> ### Author Response · Authors · 2022-11-18
> **Response to Reviewer N4rm**
>
> We thank the reviewer for the assessment and the detailed comments on our submissions.
>
> **Loss function and connection to Zhang & Chen (2021):** Implementation-wise, our training and loss are similar to Zhang & Chen (2021), but there are non-trivial differences: Zhang & Chen (2021) consider generative modeling problems, where one of the distributions is approximately Gaussian. They minimize the difference between trajectories from the forward SDE and reverse-time SDE to learn the score and drift. A subtle but important difference is that our method fits the deterministic probability flow ODE trajectories to the non-deterministic SDE paths. We provide an extended discussion of these differences now in Appendix A and show that our approach optimizes a variational lower bound on the maximum likelihood training of the CNF corresponding to the probability flow ODE. Importantly, our method optimizes the probability flow ODE to match all the marginal likelihoods \$p_t\$ for \$0 \\leq t \\leq T\$ and not only \$p_0\$ and \$p_T\$.
>
> **It is unclear why both ODE and SDE can be used. The SDE is the only correct way of sampling from the posterior** We agree with the reviewer that the correct way of sampling from the posterior is the SDE. Nonetheless, we believe an important result of our paper is to evaluate ODE solutions alongside these SDE solutions, and our results empirically show that they perform very well. We have discussed the difference in ODE vs. SDE solutions for example in section 4.1 (high reconstruction accuracy vs. fitting the data manifold). We have clarified any statements in the paper that could give the impression that we can sample from the posterior (of the reverse-time SDE) using the ODE trajectories.
>
> **Is it possible to obtain a "ground truth" posterior and compare path distributions with that?**
> It is true that the evaluations do not directly compare posterior distributions. In Table 1, we only sample one path from the reverse-time SDE for each end state we condition on. We have considered this issue extensively and agree that such a comparison would be desirable, but highly non-trivial even for simple problems. For generative modeling in computer vision, metrics like FID are typically used. They are already well-known in the community and still have shortcomings. Nonetheless, we believe that for the heat equation our proposed evaluation is a good starting point. Here, we do not compare the full trajectory, but only the starting and end points. Since the initial state cannot be fully reconstructed due to a loss of information, we simulate the state forward in time again and measure how well it matches the simulation end state using the reconstruction MSE. On the other hand, the spectral loss measures how well the initial state corresponds to the statistical properties of Gaussian random fields. Regardless of that, it is possible to compare methods that sample from the (full) posterior separately from the methods that do not in Table 1.
>
> **What is the influence of \$g=0.1\$ during training/inference?** The difference between \$g=0\$ and \$g=0.1\$ in the forward simulation is very small, since the heat equation dynamics suppress noise (we use \$g=0.1\$ for training and inference, but \$g=0\$ for calculating the reconstruction MSE). However, noise is necessary in the reverse time direction (data generation) to create smaller structures and details.
>
> **Evaluations are not precisely targeted towards posterior sampling** We have included two additional experiments, which make it possible to qualitatively compare the posterior. First, we have included an additional toy experiment in appendix B, where we compare against Implicit Score Matching. Second, we have included experiments with the 1D heat equation. We plot \$100\$ solutions of the learned reverse-time SDE condition on a simulation end state, which can be directly compared to the \$100\$ best solutions obtained by filtering a large dataset of \$10\^6\$ samples. In both cases, our method shows cleary superior performance.
>
> **Notation \$(x_t)_n \\subset \\mathbb{R}\^{d \\times T}\$. What is \$d\$?** We apologize for the unclear notation. \$d\$ refers to the data dimension and the \$t_m\$ notation should be introduced after the Euler-Maruyama discretization. We have updated the method section to clarify these aspects.
>
> **Improvements to method section and comparison to standard diffusion models**
>
> We have updated the paper with clarifications for the notation, loss function, training via the ODE solver and method section as well as highlighting the differences to standard diffusion models. An important difference is that our method fits the deterministic probability flow ODE trajectories to the non-deterministic SDE paths. We provide an in-depth discussion of these differences in appendix A.
>
> References:
>
> -   Zhang & Chen (2021): Diffusion Normalizing Flow
> -   Stachenfeld et al. (2022): Learned Coarse Models for Efficient Turbulence Simluation

---

### Official Review · Reviewer_37Sw · 2022-10-24

**Confidence:** 3
**Correctness:** 3
**Technical Novelty And Significance:** 3
**Empirical Novelty And Significance:** 3
**Recommendation:** 6

**Clarity, Quality, Novelty And Reproducibility:**

The paper is quite clear except for a few points (see above comment).

The experimental section is quite thorough and overall the paper is of good quality.

The application of diffusion models to differentiable physics is new to my knowledge.

Sufficient experimental details are given.

**Details Of Ethics Concerns:**

No ethical concern

**Strength And Weaknesses:**

STRENGTHS:
* I think the connection of differentiable physics and denoising diffusion models is interesting and is a nice application of this generative model framework to physics applications. I believe that this paper is a step towards the use of generative model for simulating trajectories in physics.
* The experimental section is quite strong. I especially enjoyed some of the findings regarding the differences between SDE and ODE which I think are both intriguing and interesting.

WEAKNESSES:
* For me the main weakness of the paper is how the training is conducted. By using the loss function the authors use in the paper they lose the flexibility of diffusion models which do not require unrolling the dynamics. If I'm right I think the authors did not apply classical diffusion models here because they did not have access to the transition kernel of the forward dynamics (since it depends on the differentiable physics operator). However there are ways to circumvent this issue by looking at the Implicit Score Matching loss, see for instance [1] (and [2] for an older reference in the score matching literature). If one does not want to use that loss then one can leverage the fact that the Euler-Maruyama discretization of the forward dynamics yield Gaussian steps (locally), see [3]. Both these approaches allow for efficient training of diffusion models even in the case where the forward dynamics is given by a differentiable physics operator. These losses are more computationally friendly and yield good results in practice

* Regarding the bidirectional optimization, recent papers on Schrodinger Bridges using diffusion models [3,4] rely on some bidirectional models. I think the authors should discuss the link between this approach and these papers. Finally, I think that the authors missed the opportunity to discuss [5] which advocates for dropping the noise in diffusion models. I think that this is not always beneficial, dropping the noise makes sense here, as the authors justify the presence of the noise by saying that it "is added to the data at each time step [and] can be regarded as either noise to the physical problem or as noise from a measurement process". Gaussian noise with such covariance seems rather arbitrary and seems to be present only for the convenience of having well-defined probability measures.

OTHER COMMENTS:
* In the reconstruction loss in the first experiment (Section 4.1). I don't really understand how this can work. If $T$ is large then the operator $P$ destroys all information and converges to a stationary state. Then how can we check that we recover $x_T$ (or are all $x_T$ the same in that case)?
* I would have appreciated more details on the bidirectional method.
* I also would have appreciated more explanations as of why SDE does not perform well in the buoyancy driven flow example.
* I am not competent enough to assess if the baselines chosen by the authors are relevant and will not comment on them.

[1] A Variational Perspective on Diffusion-Based Generative Models and Score Matching - Huang, Courville
[2] Estimation of Non-Normalized Statistical Models by Score Matching - Hyvarinen
[3] Diffusion Schrödinger Bridge with Applications to Score-Based Generative Modeling - De Bortoli, Thornton, Heng, Doucet
[4] Likelihood Training of Schrödinger Bridge using Forward-Backward SDEs Theory - Chen, Liu, Theodorou
[5] Cold Diffusion: Inverting Arbitrary Image Transforms Without Noise - Bansal, Borgnia, Chu, Li, Kazemi, Huang, Goldblum, Goldstein

**Summary Of The Paper:**

In this paper, the authors adapt the formalism of denoising diffusion models to learn physics trajectories. In particular, they consider a forward diffusion given by a differentiable physics operator. The noise of the forward diffusion is assumed to be inherent noise or measurement error. Once this is done, the authors learn the backward dynamics by backpropagating through the dynamics using the square distance between the observed forward point and generated point. They consider both Ordinary Differential Equation (ODE) and Stochastic Differential Equation (SDE) settings. They apply their method to 3 problems: a toy heat equation problem, a buoyancy-driven flow with obstacles and an isotropic turbulence problem. The method outperforms other neural network approaches to learn dynamics.

**Summary Of The Review:**

I think this paper represents a nice application of diffusion models to physics.
Apart from the training procedure which seems strange I think the methodology and experimental settings are valid.
I think this paper opens the door to more fruitful interactions between the diffusion models and physical systems.

---

> ### Author Response · Authors · 2022-11-18
> **Response to Reviewer 37Sw**
>
> We thank the reviewer for the assessment and the detailed comments on our submissions.
>
> **Losing flexibility of diffusion models and unrolling of dynamics** We’d like to point out that our formulation is very flexible in terms of allowing for different drift functions, while at the same time preventing the use of known transition kernels. We believe our results show that this trade-off pays off in the context of physics functions and unrolling the dynamics for multiple steps stabilizes the simulation for longer time horizons. However, our framework can also be used based on single step updates, i.e. unrolling the dynamics for one time step only. We give additional results for the heat equation, where we vary the number of steps for which we unroll the dynamics in Appendix C. For example, we empirically observe that increasing the rollout length from 2 to 32 improves the reconstruction MSE from 4.6e-5 to 0.8e-5 for the ODE. For the SDE longer rollouts are even more important and for a too short rollout, the reverse-time SDE trajectories are not stable (reconstruction MSE of 8.7e3 for rollout 2 vs. 9.5e-4 for rollout 32). For the reverse-time SDE, interactions between ill-conditioned physics and noise are very difficult to control when only considering single step updates during training.
>
> **Comparison to Implicit Score Matching** This is a good suggestion. We actually considered the use of Implicit Score Matching (ISM) in earlier stages of our work. We have included a comparison between ISM and our method for a simple toy problem in Appendix B. We found that while ISM is feasible, the performance of our method is clearly superior. ISM is purely data-driven and does not include the physics dynamics, whereas our method does. As a result, the trajectories of the reverse-time SDE of our method correspond to the correct distribution, whereas ISM has much more difficulty here, cf. Fig. 10.
>
> **Additional literature comparisons** We thank the reviewer for the interesting suggestions. We have included a discussion of these papers in the updated paper.
>
> **Modeling the physical system as a SDE** We agree that adding noise to the physical system and modeling it as a SDE is still uncommon and bridging the gap between models that inherently include noise and ones that are noise-free will be an important task for future research. Nonetheless, there are physical systems that can be well approximated in this framework, for example the heat equation in section 4.1. If the noise scale is small, then the dynamics of the heat dissipation, which suppress noise, remain largely unaffected by it. On the other hand, for the time-reverse direction, noise is necessary as a source for new information and diversity. There are other examples of physical systems where chaos and randomness on a small scale are an important driver of chaotic dynamics, for example turbulence simulations. In section 4.2, we take a step back from modeling the physical system as a SDE and use a deterministic simulation where noise is added afterwards as a i.i.d. Gaussian to each sampling point. This models measurement errors. Our evaluation shows that in this case our methods for ODE and SDE inference still give good results.
>
> **Reconstruction MSE for heat equation experiment:** If T is large, then the simulation end state will be very similar for all x, as most information is destroyed by the heat dissipation. However, we choose T large enough so that it destroys some information, but not all information.
>
> **Bidirectional training:** We have included more details about the method in the appendix. During bidirectional training, we alternate between training a batch of trajectories for the increasing time direction \$(0 \\to T)\$ and the decreasing time direction \$(T \\to 0)\$ with the same loss function for both.
>
> **Bad performance of SDE for the buoyancy driven flow:** We observed that when the negative time integration for the physics solver P is not sufficiently accurate and introduces approximation errors when reversing the dynamics. As a result, the physics-only version performs very poorly, as seen in e.g. Fig. 21, and the score network learns to represent the score plus additional corrections to the physics dynamics. For example, the score network \$s_\\theta\$ has to correct the transportation within the marker field, which should mainly be prescribed by the physics. For the ODE version, this is not a problem. However, for the SDE, we increase the multiplicative factor before \$s_\\theta\$ and add additional noise during inference. Then, the corrections to the physics by \$s_\\theta\$ can overshoot, which makes it more difficult during inference to produce accurate trajectories. We believe that this behavior could be mitigated by an improved reversible time integrator for the physics solver and smaller time steps.
>
> References:
> -   Zhang & Chen (2021): Diffusion Normalizing Flow [(https://arxiv.org/abs/2110.07579)](https://arxiv.org/abs/2110.07579)

---

### Author Response · Authors · 2022-11-18
**Response to all reviewers**

We would like to thank all reviewers for their assessment and constructive comments. We also appreciate more critical comments that point out open questions to the relation of our method with the score matching objective. As a consequence, we provide additional theoretical justification as well as including direct comparisons of the learned score with established score matching methods as an additional empirical verification of our method. We have posted a revision of our submission with changes highlighted in blue color. In particular, our revised version includes the following major updates:

-   We have clarified the method section and include additional implementation details. We also give new theoretical justification on how our proposed method can be understood as maximizing a variational lower bound for the maximum likelihood training of the continuous normalizing flow (CNF) of the probability flow ODE. Our derivation in appendix A also explains how maximum likelihood training can be used for minimizing the score matching objective.
-   We additionally included a new comparison of Implicit Score Matching (ISM) and our method in appendix B. This allows for a direct comparison of the learned score between ISM and our method, which validates the methodology as well as highlights the importance of including the physics dynamics in the training.
-   We include additional experiments for the heat equation in Appendix C, where we vary the rollout length from very short (local) rollouts to rollouts spanning the entire simulation trajectory. Our evaluation shows that for physical systems, longer rollouts are important for stable training due to interaction effects between ill-conditioned physics and noise.
-   We include an additional evaluation of the posterior of the reverse-time SDE trajectories for the heat equation in appendix D, which demonstrates the high quality and diversity of the reverse-time SDE trajectories.

We give more detailed answers to the comments by each reviewer below.

---

### Decision · Program_Chairs · 2023-01-20

**Decision:**

Reject

**Justification For Why Not Higher Score:**

Please see critical weaknesses listed above.

**Justification For Why Not Lower Score:**

N/A

**Metareview: Summary, Strengths And Weaknesses:**

Summary: The goal of this paper is to  model the distribution of initial macroscopic states for a given end state of a physical system, for which the dynamics is given by a differentiable simulator. This is done by setting up a Neural ODE L2 training objective to learn the so-called Probability Flow ODE, and then sampling from the associated reverse time SDE. The method is demonstrated on a toy heat equation problem, a buoyancy-driven flow with obstacles and an isotropic turbulence problem.

Strengths: Incorporating a strong prior in the form of a differentiable simulator in score function learning appears to be a novel way to formulate inverse problems.  In the experiments, the proposed model improves over directly learning an unstructured neural network highlighting the role of priors, or simply running the simulator backwards which does not capture stochasticity.

Weaknesses:  The discussion around weaknesses in the review process ended up converging around clarity  --  the paper makes several claims about the relationship between this model and SDE-based score-matching diffusion generative models, omitting necessary details about the training procedure and its connections to approximate score matching
 / max-likelihood training (e.g. the dependence on the window size are not very well explained in the paper).  The lack of sufficient rigor makes it harder to appreciate the mechanisms that allow the method to do well and reduces the confidence in reproducibility.